# Reward-related choices determine information timing and flow across macaque lateral prefrontal cortex

Hua Tang [1], Ramon Bartolo [1] & Bruno B. Averbeck [1✉]

Prefrontal cortex is critical for cognition. Although much is known about the representation of cognitive variables in the prefrontal cortex, much less is known about the spatio-temporal neural dynamics that underlie cognitive operations. In the present study, we examined information timing and flow across the lateral prefrontal cortex (LPFC), while monkeys carried out a two-armed bandit reinforcement learning task in which they had to learn to select rewarding actions or rewarding objects. When we analyzed signals independently within subregions of the LPFC, we found a task-specific, caudo-rostral gradient in the strength and timing of signals related to chosen objects and chosen actions. In addition, when we characterized information flow among subregions, we found that information flow from action to object representations was stronger from the dorsal to ventral LPFC, and information flow from object to action representations was stronger from the ventral to dorsal LPFC. The object to action effects were more pronounced in object blocks, and also reflected learning specifically in these blocks. These results suggest anatomical segregation followed by the rapid integration of information within the LPFC.

[1] Laboratory of Neuropsychology, National Institute of Mental Health, National Institutes of Health, Bethesda, MD, USA. ✉email: averbeckbb@mail.nih.gov

Prefrontal cortex, particularly the LPFC, supports multiple cognitive operations that require flexible mappings between stimuli and actions to obtain rewards or avoid punishments[1–3]. Numerous studies have documented significant representations of the stimuli, actions, rules, and rewards that are important for these processes in LPFC population activity. In addition, lesions of the LPFC lead to deficits in cognitive operations, including working memory, sequential planning, and rule learning[4].

Although LPFC is sometimes treated as a monolithic structure, proposals have been put forward for specific functional domains within the LPFC along both ventro-dorsal and caudo-rostral axes. The pattern of anatomical connections[5,6], neurophysiological, and neuroimaging findings has suggested a "domain-specific" organization along the ventro-dorsal axis of LPFC. Support[7–10] for this proposal derives from studies that show that spatial stimuli recruit the caudal dorsolateral prefrontal cortex (cdlPFC) and object or verbal stimuli recruit the ventrolateral prefrontal cortex (vlPFC). Other studies, however, favor a model in which individual neurons that integrate different types of information are distributed throughout the LPFC. In this model, neuronal responses are shaped by cognitive demands imposed by the task rather than selectivity for specific domains[1,11]. Human imaging studies have supported both proposals[12]. Some studies are consistent with specialized processing in the dorsal and ventral subdivisions[13,14], whereas others support a generalized organization around cognitive operations rather than information domains[15].

Other groups have suggested that the frontal cortex, from the premotor cortex to the frontal pole, is hierarchically organized along a caudo-rostral axis. The specific proposals have suggested that locations along this axis relate to the level of abstraction involved in the behavioral process or the ability to temporally organize and initiate sequential behavior[16,17]. Evidence for a hierarchical organization of neural processing has been provided by functional magnetic resonance imaging (fMRI) studies[18–20] and is also supported by lesion studies[21,22]. Consistent with this, the rostral dorsolateral prefrontal cortex (rdlPFC) exhibits the largest receptive fields, longest response latencies, and the least information about stimuli, which suggests highly abstracted representations[23,24]. In contrast to the studies that suggest a caudo-rostral organization, however, recent evidence suggests that the apex of the prefrontal hierarchy resides in the middle LPFC rather than the rdlPFC[25–27]. The extent to which the LPFC is organized along a rostro-caudal axis hence constitutes a matter of debate.

In the present study, we examined signal timing and information flow, in caudo-rostral and ventro-dorsal axes in LPFC, while monkeys carried out a two-armed bandit reinforcement learning task. In the task, the animals had to either learn which of two objects was more frequently rewarded, or which of two actions was more frequently rewarded[28,29]. While the monkeys carried out the task, we recorded activity from large populations of single neurons, using eight Utah arrays. We found a substantial caudo-rostral gradient in the strength and timing of signals, relative to both chosen objects and actions. When we directly examined information flow, we found both caudo-rostral and dorso-ventral information flow that was task-specific, reflecting the cognitive process of identifying the location of a valuable object, and directing an eye movement to that location.

## Results

Two rhesus monkeys learned to perform a two-armed bandit reversal learning task with a stochastic reward schedule (Fig. 1a, b). The task featured two types of learning blocks: object-based (What) and location or action-based (Where). The monkeys were tested on multiple, randomly interleaved blocks each session. Each block was either a What block or a Where block. In addition, the options were stochastically rewarded using a 70%/30% reward schedule. At the beginning of each block, the monkeys were presented with two novel objects as choice options. The monkeys selected one option per trial by making a saccade and fixating on their choice. The individual stimuli were randomly assigned to the left or right of fixation on each trial. In What blocks, the higher-probability choice was one of the two objects independent of the action needed to select it. In Where blocks, the higher-probability choice was one of the two actions independent of the object at the target location. There were no explicit cues to indicate the block type before the start of each trial. The monkeys determined the block type through inference over choices and feedback. In each block, on a randomly chosen trial between 30 and 50, the reward mappings were reversed, making the previously less rewarded option the more rewarded option and vice versa. The monkeys had to detect the reversal and switch to choosing the other option. The block type never switched across reversals. After the 80 trials had been completed, a new block began, and two novel objects were introduced. The monkeys then had to learn again via trial and error whether the reward mapping was based on the chosen action (left or right saccade) or the chosen object.

**Choice behavior**. Monkeys completed 24 blocks per session. We visualized the monkeys' choice behavior by aligning each block around the reversal trial before averaging (Fig. 1c). Monkeys learned to select the better option during the acquisition phase and to switch their choice behavior when they detected the contingency reversal. The fraction of correct choices reached about 80% for both What and Where blocks after learning. We analyzed the reaction times (RTs) in both What and Where blocks. In What blocks, the average RT was 216.8 ms (SD = 12.3 ms), and in Where blocks, the average RT was 205.2 ms (SD = 19.6 ms). These RTs differed by block type (paired $t$ test, $t$ (7) = 4.08, $p$ = 0.005).

**Neural encoding of chosen action and object across arrays**. Neural activity was recorded from eight arrays implanted in each animal, four in the left hemisphere and four in the right (768 total electrodes), in the corresponding locations across hemispheres and monkeys (Fig. 1d). These arrays were located in the rostral dorsal (rdlPFC), middle dorsal (mdlPFC), caudal dorsal (cdlPFC), and ventral LPFC (vlPFC). For each monkey, four sessions of neurophysiology data were analyzed, in which we recorded the activity of 3443 neurons (Supplementary Table 1) from monkey V (877, 942, 1026, and 598 for each session) and 2689 neurons from monkey W (680, 747, 677, and 585 for each session).

A large proportion of recorded neurons responded to the task. Broad diversity of activity profiles was observed, including differential responses to both chosen objects and locations (Supplementary Fig. 1). The population firing rate of the neurons that responded during the task decreased along the caudo-rostral axis. Although the neural populations tended to show a stronger response to contralateral stimuli, their overall responses to different options (left vs. right, object A vs. object B) were similar (Supplementary Fig. 2).

We began by characterizing single-cell encoding of the chosen object and the action. This analysis was split out by block type and anatomical location (Fig. 2; Supplementary Fig. 3). To be more specific, this analysis was performed to test whether and when single cells discriminated between chosen and nonchosen options. This analysis does not specifically assess whether neurons encoded the object identity or location, only differences between chosen and

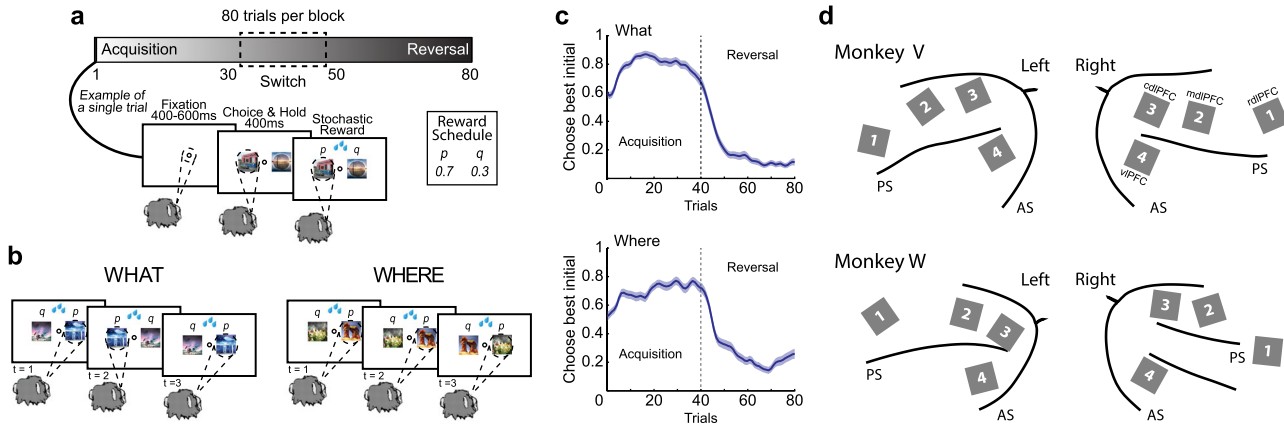

**Fig. 1 Reversal learning task, behavior, and array maps. a** Block structure of the task and single-trial behavior. In each trial, the animals first acquire the central fixation. Two objects are then shown on the left and right sides of the fixation dot. The animals make an eye movement to acquire one of the objects/locations and are then rewarded based on the block type and the reward probability assigned to that object/location. **b** Each block is either a What block, where rewards are associated with objects or a Where block, where rewards are associated with locations. The block type is not indicated and it is randomly interleaved. **c** Choice behavior for two monkeys. The fraction of times that the monkeys chose the initially higher rewarded visual stimulus option in What and Where blocks. Solid lines show the means, and the shaded regions show the mean ± SEM, which were computed across all sessions. The vertical dashed line shows the average reversal point. Data from two monkeys, $n = 8$ sessions. **d** Schematic of array locations. Four Utah arrays (indicated by gray squares) were implanted in the LPFC of each hemisphere for each monkey. Array 1 was located in the rostral dorsal LPFC (rdlPFC), array 2 was located in the middle dorsal LPFC (mdlPFC), array 3 was located in the caudal dorsal LPFC (cdlPFC), and array 4 was located in the ventral LPFC (vlPFC). Monkey W had an unusual sulcal pattern in the right hemisphere. AS arcuate sulcus, PS principal sulcus. Source data are provided as a Source Data file.

unchosen actions, or objects. We examined the encoding of task variables during the initial hold period and after the object was presented. Following object onset (Fig. 2g), there was a consistent caudo-rostral gradient, such that more neurons were task-responsive in the vlPFC and cdlPFC than in the mdlPFC and rdlPFC (Array; $F (3, 237) = 96.72, p < 0.001$). The gradient was more pronounced for object encoding than for action encoding (Array × Domain; $F (3, 237) = 24.83, p < 0.001$). We also carried out planned comparisons between the vlPFC and cdlPFC to examine the hypothesis that there was a ventro-dorsal gradient in object vs. action representation in the caudal LPFC. When we examined the representation of objects vs. actions between the cdlPFC and vlPFC, we found a significant interaction (Array × Domain; $F (1, 28) = 10.08, p = 0.004$). Post hoc comparisons showed that there were more neurons encoding the chosen object in the vlPFC than cdlPFC ($t (7) = 6.2, p < 0.001$) and a trend for significantly more cells to encode chosen actions in the cdlPFC than the vlPFC ($t (7) = 2.4, p = 0.044$).

Encoding during the hold period before the options were presented also reflected learning. The best option in each block was represented in the neural activity during the baseline hold period. There was a stronger encoding of objects in What vs. Where blocks relative to actions in Where vs. What blocks (Fig. 2c, f; $F (1, 237) = 89.13, p < 0.001$). Thus, the neural activity reflected planned choices, following learning, during the hold period.

Next, we examined response latency. For both chosen actions and objects across both block types, the response latency increased from caudal to rostral (Supplementary Table 3; Fig. 2h, i; Array; $F (3, 232) = 117.25, p < 0.001$). Latencies for chosen objects were shorter than latencies for chosen actions (Domain; $F (1, 232) = 50.8, p < 0.001$). Latencies for chosen objects preceded object onset in What blocks in the cdlPFC and vlPFC (Fig. 2i; paired $t$ test, $t (7) = 11.3, p < 0.001$ for the cdlPFC; $t (7) = 19.0, p < 0.001$ for the vlPFC). (Note that ANOVAs were conducted across the entire block, and animals likely explored both What and Where strategies at the beginning of the block[28], leading to the encoding of objects in Where blocks prior to object onset). As noted above, however, the encoding of objects was weaker in

Where blocks than What blocks during the hold period. Responses also tended to be earlier for chosen objects in What blocks than Where blocks (Block type; $F (1, 96) = 234.54, p < 0.001$), but they were not, on average, shorter for chosen actions in Where blocks than What blocks (Block type; $F (1, 106) = 1.30, p = 0.256$). The onset latency for the chosen action was, however, shorter in the mdlPFC in Where blocks than What blocks (paired $t$ test, $t (7) = 10.11, p < 0.001$). Therefore, there was a caudo-rostral gradient in response latencies, and the latencies reflected the relevant choice domain in the corresponding blocks.

We also examined the encoding of the block type at a single-cell level (Supplementary Fig. 4). The block type is an abstract rule that defines the relevant choice dimension. When we examined block type, we found that it was more strongly represented in caudal areas (Following cue period, Array; $F (3, 56) = 12.14, p < 0.001$), similar to the other factors. Therefore, we did not find an enhanced representation of block type in more anterior parts of LPFC.

We were also interested in whether the same group of neurons tended to respond within the same domain (i.e., chosen action vs. object across block type) or within the same task condition (i.e., responses to action and object but confined to What or Where blocks). Therefore, we also examined the co-occurrence of encoding in several ways. First, we examined the co-occurrence of action and object encoding within each block type (Supplementary Fig. 5a, b). Second, we examined the co-occurrence of action (Supplementary Fig. 5e) and object encoding (Supplementary Fig. 5f) in both What and Where blocks. Finally, we also examined cross-domain encoding of action in What blocks and object in Where blocks (Supplementary Fig. 5c) and object encoding in What blocks and action encoding in Where blocks (Supplementary Fig. 5d). Similar to the encoding of single variables, the co-occurrence of multiple variables tended to be stronger in the vlPFC and cdlPFC (Supplementary Fig. 5g; Array; $F (3, 371) = 133.69, p < 0.001$). Interestingly, we found that neurons tended to encode the same domain across task conditions (Supplementary Fig. 5h; $F (1, 371) = 25.37, p < 0.001$). Specifically, there was a stronger co-occurrence (i.e., neurons significant in both conditions) of action encoding across

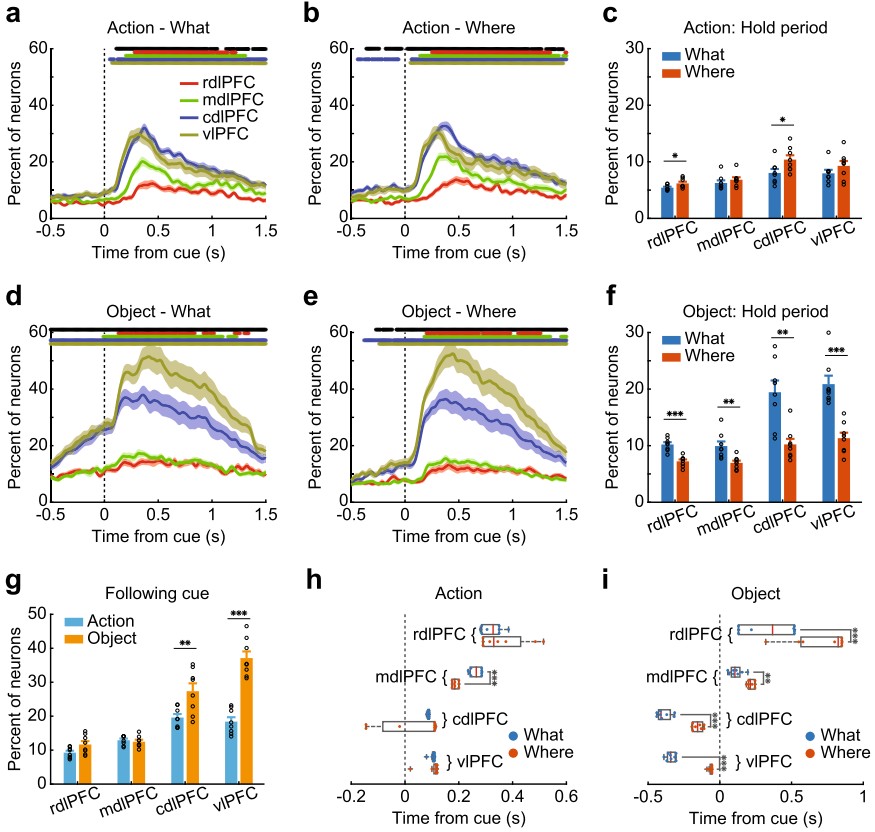

**Fig. 2 Population encoding of chosen action and object. a, b, d, e** Percentage of task-related neurons in each region that encoded actions in What (**a**) or Where blocks (**b**), encoded object identity in What (**d**) or Where (**e**) blocks. **c, f** The average percentage of task-related neurons in each region during the initial hold period, encoding action (**c**) or object information (**f**). Hold period: −0.5 to 0 s from cue onset. **g** The percentage of task-related neurons in each region by domain type, averaged from 0 to 1.5 s from cue onset. **h, i** The response latency of the neuronal populations in each region for encoding action (**h**) and object identity (**i**), split by block type. The solid circles represent the response latency of each session. Boxplot box indicates the first and third quartile, the center line of the box indicates the median, and whisker lengths reflect the interquartile range multiplied by 1.5. Shaded zones and error bars represent mean ± SEM, $n = 8$ for each line, bar, or box. A two-sided $t$ test was used to compare two populations, $*p < 0.05$, $**p < 0.01$, $***p < 0.001$. The black * symbols at the top of each panel indicate a significant difference among the four regions (one-way ANOVA, $p < 0.01$). The colored * symbols indicate a significant difference (two-sided paired $t$ test, $p < 0.01$) of task-related neuron percentage between the corresponding region and its baseline. Source data are provided as a Source Data file.

What and Where blocks (Supplementary Fig. 5e) and object across What and Where blocks (Supplementary Fig. 5f), when compared to action and object in What blocks (Supplementary Fig. 5a), or action and object in Where blocks (Supplementary Fig. 5b). Cross-domain cross-block encoding (Supplementary Fig. 5c, d), however, was similar to cross-domain within block encoding (Supplementary Fig. 5a, b).

Taken together, these results suggest that there is an association between the neuronal population location in the LPFC and the response to the chosen action and object. In general, the neuronal population in the caudal LPFC showed stronger encoding, and co-occurrence rate, a shorter response latency, and a stronger response to object vs. action information than the rostral LPFC neuronal populations.

**Decoding of chosen actions and objects from neural activity.** The encoding analysis addressed how individual neurons respond to chosen objects and actions. To further understand how the neural populations coded object and action information, we carried out a decoding analysis, using all neurons simultaneously recorded within each array to predict either the chosen action or the object (Fig. 3; Supplementary Fig. 6). The results were generally consistent with the encoding analysis (Fig. 2), although there were some differences. We again found increased decoding performance in the cdlPFC and vlPFC (Fig. 3g; Array; $F$

$(3, 237) = 44.07$, $p < 0.001$), compared to the rdlPFC and mdlPFC. However, the decoding analysis showed that there was more information about chosen actions than chosen objects (Domain; $F$ $(1, 237) = 100.6$, $p < 0.001$), unlike what we found for encoding at the single-cell level (Fig. 2g). This suggests that single neurons, when aggregated into a population, contain more information about chosen actions than objects, even though more neurons encode objects than actions. We also examined decoding during the hold period, before the options were presented, and found a stronger representation of actions in Where vs. What blocks (Fig. 3c) relative to objects in What vs. Where blocks (Fig. 3f; Block type × Domain; $F$ $(1, 237) = 50.23$, $p < 0.001$). Therefore, when the animals learned the best choice in each block, the choice was represented before the options were presented.

Similar to the encoding analysis, for both chosen actions and objects across both block types, the information latency increased from caudal to rostral (Fig. 3h, i; Array; $F$ $(3, 182) = 119.30$, $p < 0.001$). Latencies for chosen objects were also shorter than latencies for chosen actions (Domain; $F$ $(1, 182) = 45.14$, $p < 0.001$). Latencies for chosen objects preceded object onset in What blocks. Responses also tended to be earlier for chosen objects in What blocks than Where blocks in the cdlPFC and vlPFC (Fig. 3i, paired $t$ test, $t$ $(7) = 16.71$, $p < 0.001$ for the cdlPFC, $t$ $(7) = 6.45$, $p < 0.001$ for the vlPFC) and for chosen actions in Where blocks than What blocks (Block type; $F$ $(1, 109) = 37.22$,

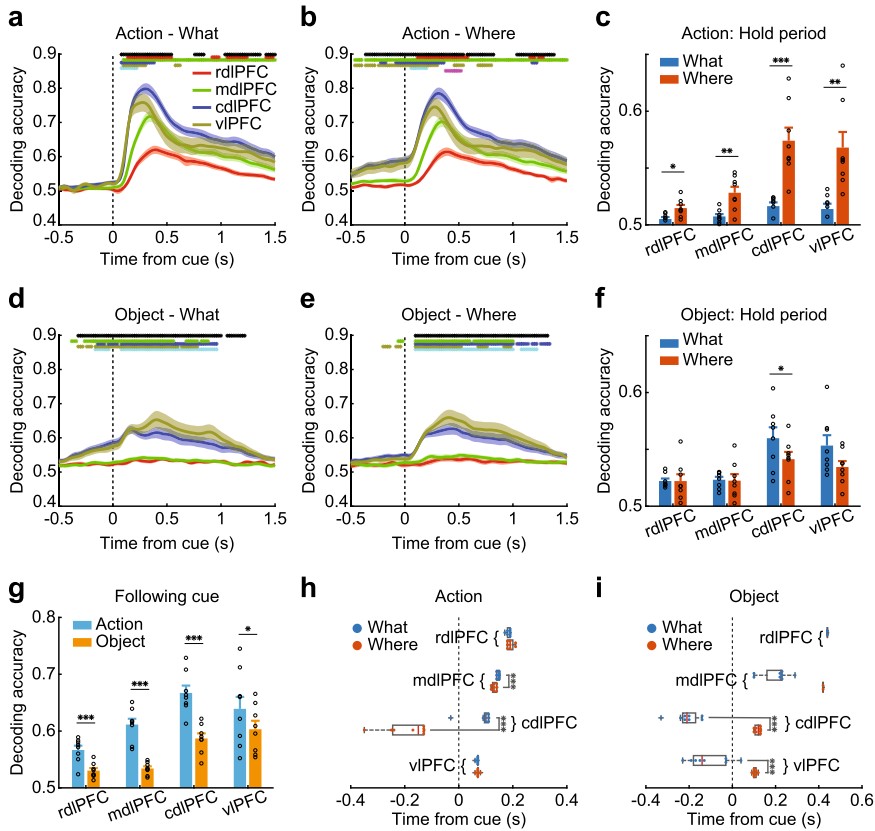

**Fig. 3 Decoding of chosen action and object. a**, **b**, **d**, **e** The time course of decoding accuracy in each region that encoded action in What (**a**) or Where blocks (**b**), or encoded object in What (**d**) or Where (**e**) blocks. **c**, **f** The average hold period decoding accuracy in each region, when decoding action (**c**) or object information (**f**). Hold period: −0.5 to 0 s from cue onset. **g** The decoding accuracy in each region split by domain type, averaged from 0 to 1.5 s from cue onset. **h**, **i** The response latency of decoding accuracy in each region for encoding action (**h**) and object (**i**), split by block type. The solid circles represent the response latency of each session. Boxplot box indicates the first and third quartile, the centerline of the box indicates the median, and whisker lengths reflect the interquartile range multiplied by 1.5. Shaded zones and error bars represent mean ± SEM, $n = 8$ for each line, bar, or box. A two-sided $t$ test was used to compare two populations, *$p < 0.05$, **$p < 0.01$, ***$p < 0.001$. The black * symbols at the top of each panel indicate a significant difference among the four regions (1-way ANOVA, $p < 0.01$). The red, green, blue, yellow, cyan, and magenta * symbols indicate a significant difference (two-sided paired $t$ test, $p < 0.01$) between the rdlPFC and mdlPFC, rdlPFC and dlPFC, rdlPFC and vlPFC, mdlPFC and cdlPFC, mdlPFC and vlPFC, and cdlPFC and vlPFC. Source data are provided as a Source Data file.

$p < 0.001$). Therefore, there was a caudo-rostral gradient in response latencies, and the latencies reflected the relevant choice domain in the corresponding blocks.

**Decoding of reward from neural activity**. We also decoded the reward outcome (Supplementary Figs. 7 and 8) for comparison with the choice variables. We found substantial information about the outcome that differed across arrays (Supplementary Fig. 7c; Array; $F (3, 115) = 15.13, p < 0.001$). The reward did not, however, differ across block type (Block type; $F (1, 115) = 0.03, p = 0.856$). The onset latencies were also consistent across arrays (Supplementary Table 4, Supplementary Fig. 7d, Array; $F (3, 10) = 1.12$, $p = 0.386$). Across arrays, the decoding accuracy of reward was higher than the decoding accuracy averaged across action and object identity (Fig. 3g and Supplementary Fig. 7c; Reward vs. choice; $F (1, 59) = 26.44, p < 0.001$), especially for the rdlPFC and mdlPFC.

**Prediction of action and object identity**. The analyses above revealed that neuronal populations along the caudo-rostral axis of LPFC encoded both chosen actions and objects. In addition, the strength of the signal and the onset latency varied from caudal to rostral. To characterize the flow of information along the caudo-

rostral axis, we next examined trial-by-trial directed information flow among arrays (Fig. 4). Specifically, we asked whether the signal in one array could be predicted by the signals in the other arrays and whether this prediction would be directed (i.e., caudal to rostral) and task-dependent.

To begin, we calculated the posterior probability, using the decoding model, of the chosen action or object (Fig. 4a–c), given the neural activity in 20-ms bins on each array (Fig. 4d, e). This analysis resulted in a time series that represented the information (i.e., the posterior probability given the neural activity) about each choice, at each point in time, on each array (Fig. 4c; here we show only 3 arrays for simplicity, but all arrays were used in the full analysis). We sought to characterize the flow of this information across arrays. We did this using a Granger Causal modeling framework. Specifically, can the future information on an array be predicted with the current and past information on other arrays, after accounting for future predictions with the same array? We worked with information, instead of spikes, because increases in information can be represented as increases or decreases in firing rates. Information is, of course, just a processed version of spikes and working with information, therefore, maps the population neural activity into the space relevant for behavior. After computing posteriors on each array for each trial, we fit a Granger model, which predicted the posterior on one array

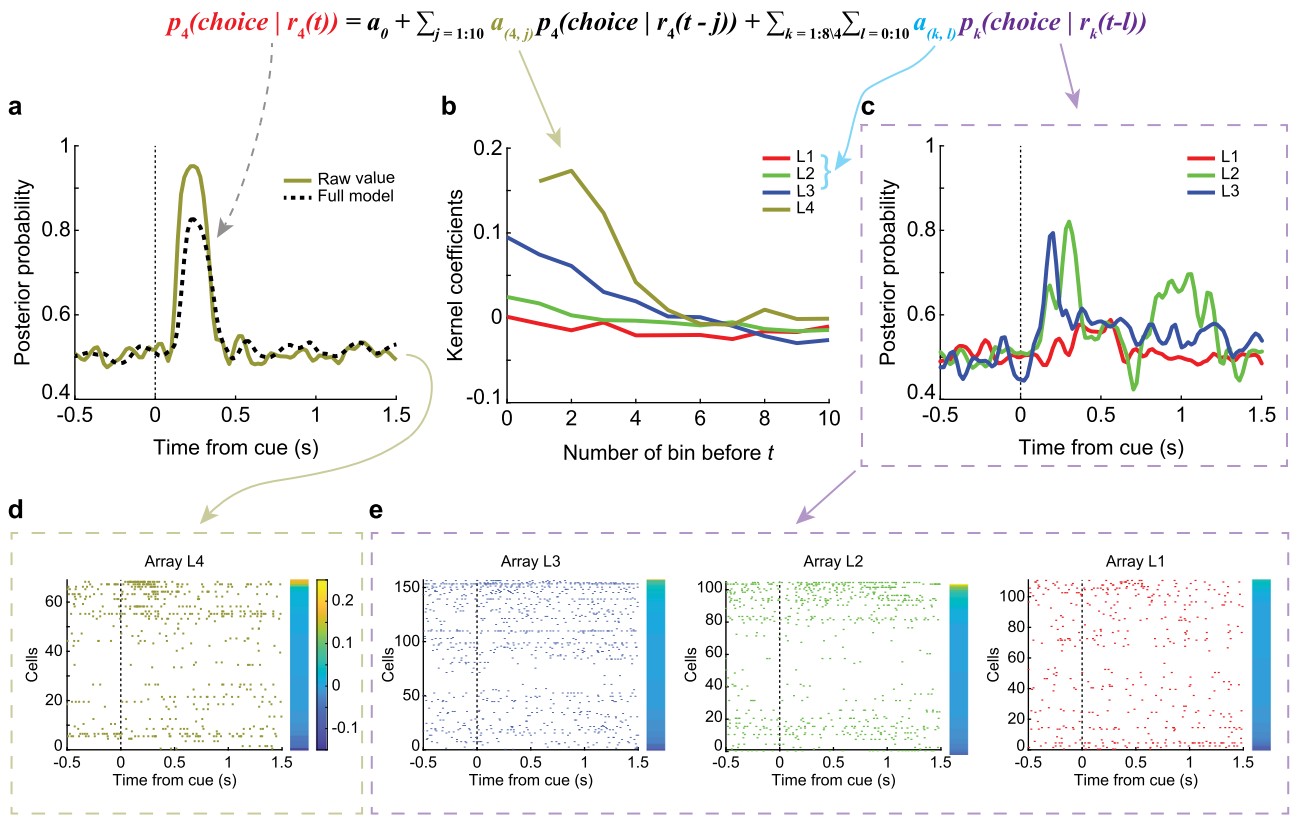

$$p_4(choice \mid r_4(t)) = a_0 + \sum_{j=1:10} a_{(4,j)} p_4(choice \mid r_4(t-j)) + \sum_{k=1:8\backslash 4} \sum_{l=0:10} a_{(k,l)} p_k(choice \mid r_k(t-l))$$

**Fig. 4 Information-transfer model.** A single-trial example of the information-transfer model. Predicting the decoding accuracy of action in Where blocks of array 4 in the left hemisphere (array L4) with the input from all eight arrays. **a** The posterior probability of the output array: the raw and the predicted value. **b** The kernel coefficients of four input arrays (arrays 1–4 in the left hemisphere). **c** The posterior probability of the input arrays (only showing arrays L1–L3). **d**, **e** The raster of all neurons in the output (**d**) and input arrays (**e**). Ordered by the difference of distance to two choices, which was indicated by the color bars.

(e.g., array L4, Fig. 4a) using the posteriors on the other seven arrays, and lagged values of the posterior on the same array. We refer to this as the Full model as it includes all predictors we measured. The model resulted in a set of kernel coefficients (Fig. 4b), which were convolved with the posteriors in the input arrays (Fig. 4c) to generate a prediction of the posterior on the output array (Fig. 4a). The kernel coefficients show the effect of lagged information in one area on future information in another area. This example shows that prediction tended to be the strongest at short delays, and decays with time (Fig. 4b). Across the caudo-rostral axis, we found that the posteriors in the output arrays could be well predicted using the Full model (Fig. 5c–f). Given the large amount of data, the full regressions were always significant ($p < 0.01$). This was consistent across arrays, even though the average posterior probabilities were higher for the caudal arrays (Fig. 5e, f) than the rostral arrays (Fig. 5c, d). The higher posteriors are consistent with the increased decoding performance in the caudal arrays.

To examine the contribution of individual arrays to the Full model and to characterize information flow, we dropped each individual array from the model and recomputed predictions (Supplementary Fig. 9). To simplify these results, we examined the average effects of dropping the arrays at the corresponding locations in the left and right hemispheres (Fig. 5). For example, when predicting the posterior in array 4 (cvlPFC) in the left hemisphere, we dropped array 3 (dlPFC) in the left/right hemisphere and averaged the predictions (Fig. 5a, drop-L/R3 Partial model). Next, we calculated the difference in the predicted posterior ($\Delta$Posterior) between the Full model and the drop-L/R3 Partial model (Fig. 5b). This estimated the partial contribution of

the cdlPFC (bilaterally) to the posterior probability in the vlPFC. Specifically, this estimates the Granger causal influence of past activity in the cdlPFC to future activity in the vlPFC, and therefore, assesses the relative flow of information from the cdlPFC to the vlPFC.

This analysis showed a multiphasic contribution of the cdlPFC, peaking at around 200 ms after object onset, to the signal in the vlPFC. Across arrays, we found that inputs from neighboring arrays tended to play a large role. For example, the contribution from the cdlPFC to the vlPFC was relatively large (Fig. 5f, j), as was the contribution from the vlPFC to the cdlPFC (Fig. 5e, i) and the rdlPFC and cdlPFC to the mdlPFC (Fig. 5d, h). These effects were consistent in both hemispheres (Supplementary Fig. 9).

We next summarized these effects by calculating the fraction of variance about future information predicted in each array by the other arrays (Fig. 6). This was done in the same way as the analysis above (Fig. 5). We calculated the fraction of variance in the posterior explained by the Full model, and then dropped an array from the model, and recomputed the fraction of variance explained by the Partial model. The difference in variance explained between the Partial and Full models, normalized by the variance explained in the Full model, characterized the partial contribution of each array to the other arrays, and therefore, the flow of information from one array to another.

We split this analysis out by several factors, which allowed us to test specific hypotheses statistically. First, the ordinal distance between arrays (see Fig. 1d for an ordinal number of arrays), from the ventro-caudal array along the dorso-rostral axis (although we also show data without collapsing by ordinal distance in Supplementary Fig. 10). This allowed us to see if information

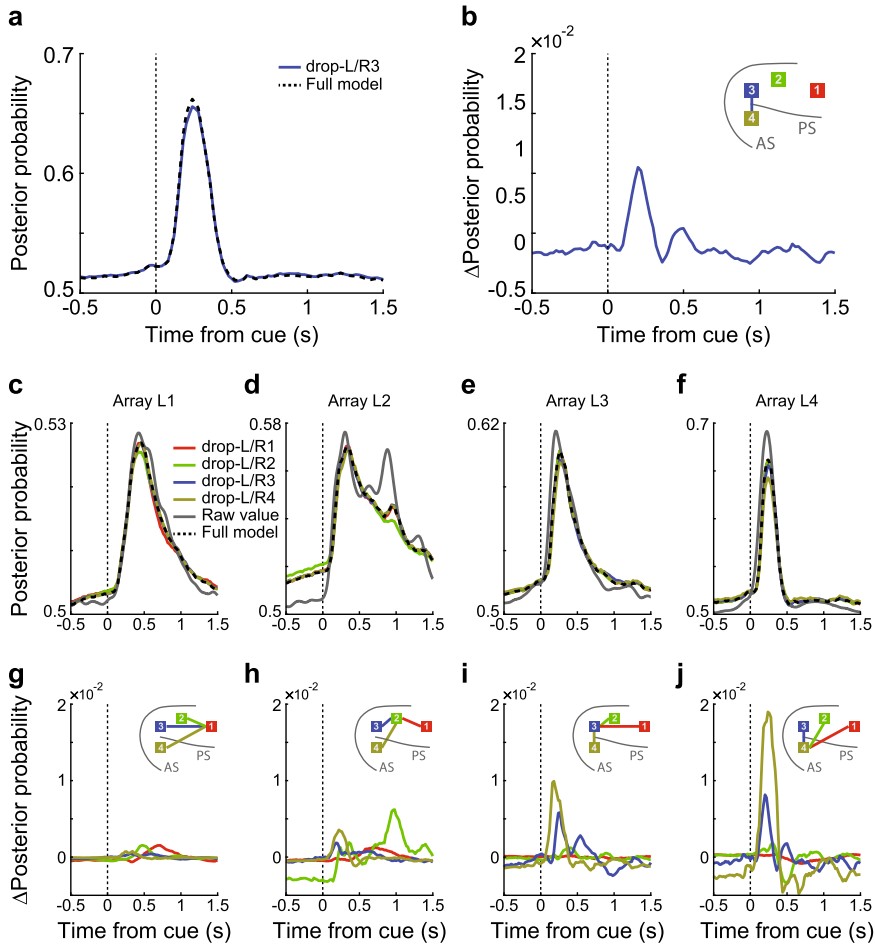

**Fig. 5 Modeling example.** An example of the posterior probability of the information-transfer model, predicting the decoding accuracy of action in Where blocks of array 1 (**c, g**), array 2 (**d, h**), array 3 (**e, i**), and array 4 (**f, j**) in the left hemisphere with input from all eight arrays. **a** The predictions of array L4 from the Full model and the drop-L/R3 Partial model. **b** The difference between the Full model and the drop-L/R3 Partial model. Inset: the colored line connects the dropped input array (indicated by the color of the line) and the output array. Here, array 3 as an input array, and array 4 as the output array. **c–f** The posterior probability of the raw value, the Full model, and the Partial models with the average effect of the arrays at the corresponding locations in the left and right hemispheres. **g–j** The difference of posterior probability between the Partial models and the Full model. Inset: the colored lines connect the dropped input (indicated by the lines' color) and the output array.

flow tends to be stronger locally or caudal vs. rostral. Second, we split the analysis by block type to see if information flow depended on whether the animal was learning to select actions or objects. The third factor was the prediction of either chosen actions or objects (Fig. 6; Supplementary Fig. 11). We found that there was more information flow for chosen actions than chosen objects (Fig. 6a vs. b; $F (1, 1517) = 20.21$, $p < 0.001$), consistent with the increased information about actions relative to objects at the population level (Fig. 3g). Connectivity within the LPFC is recurrent[30], and therefore information will flow in both directions. However, we considered whether there was more information flow in caudo-rostral vs. rostro-caudal directions. We found that there was stronger flow in the caudo-rostral direction than in the rostro-caudal direction when predicting actions (Fig. 6a; unpaired $t$ test, $t (766) = 2.76$, $p = 0.006$) but not objects (Fig. 6b). There was also stronger information flow between adjacent arrays (Fig. 6a; unpaired $t$ test, $t (766) = 7.59$, $p < 0.001$; Fig. 6b; unpaired $t$ test, $t (766) = 11.38$, $p < 0.001$).

**Prediction of chosen action and object across domain types.** Next, we examined whether information about actions could be used to predict information about objects and vice versa. In Where blocks, the animals did not have to use object information to select

an action, they could simply preplan an action. The action was directed at an object. However, in What blocks, the animals had to use object information to find the object, and then direct a saccade toward it. Therefore, we expected information flow from object to action, but less flow from action to object (Supplementary Fig. 10). We found that there was stronger flow in the caudo-rostral direction than in the rostro-caudal direction when predicting action with an object (Fig. 6c; unpaired $t$ test, $t (766) = 7.39$, $p < 0.001$). We also found that there was stronger flow in the rostro-caudal direction than in the caudo-rostral direction when predicting object with action (Fig. 6d; unpaired $t$ test, $t (766) = 3.83$, $p < 0.001$). We repeated these analyses using only lagged values of information in the arrays used for prediction and found highly consistent results (i.e., the $l$-variable index from 1 to 10 instead of 0 to 10; Supplementary Fig. 12a, predicted actions with objects; unpaired $t$ test, $t (766) = 7.45$, $p < 0.001$. Supplementary Fig. 12b, predicted objects with actions; unpaired $t$ test, $t (766) = 3.93$, $p < 0.001$). We also found that there was increased information flow from object to action in What blocks compared to Where blocks (Fig. 6c; Block type; $F (1, 756) = 5.1$, $p = 0.024$). There was no difference in information flow from actions to objects across block types (Fig. 6d; Block type; $F (1, 756) = 0.4$, $p = 0.544$). In addition, the information flow from objects to

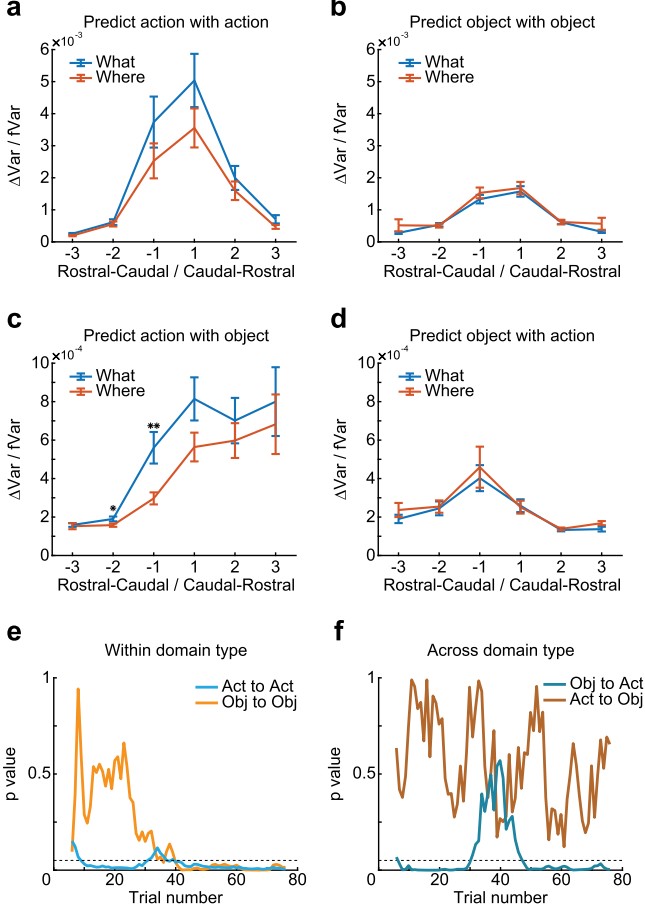

**Fig. 6 Prediction of chosen action and object. a–d** Contribution of arrays separated by different ordinal distances either rostral or caudal to each array to the prediction of posterior in each array. The *x*-axis indicates the ordinal distance (see Fig. 1d for array number) of the arrays that were dropped in the partial model. The *y*-axis indicates the difference of variance explained between the Partial models (i.e., when dropping one array, bilaterally) and the Full model (ΔVar), normalized by the variance explained in the Full model (fVar). Larger values indicate a stronger prediction of the posterior from the arrays separated by distances indicated on the *x*-axis. The separate panels show the prediction of decoding accuracy of action with action (**a**), predicting object with an object (**b**), predicting action with an object (**c**), and predicting object with action (**d**). Rostral–caudal indicate information flow from rostral to caudal LPFC, labeled by negative ordinal distance; Caudal–rostral indicates information flow from caudal to rostral LPFC, labeled by positive ordinal distance. Error bars represent mean ± SEM, n = 32/64/96/96/64/32 for the ordinal distance of −3/−2/−1/1/ 2/3. A two-sided *t* test was used to compare two populations, *p = 0.0383, **p = 0.0032. **e**, **f** The *p* value of the difference in information flow between tasks (What and Where), aligned by the trial index, when doing predictions within (**e**) or across (**f**) domain type. Bin = 10 trials, step = 1 trial. Dash lines represent *p* = 0.05, Act represents action, Obj represents an object. Source data are provided as a Source Data file.

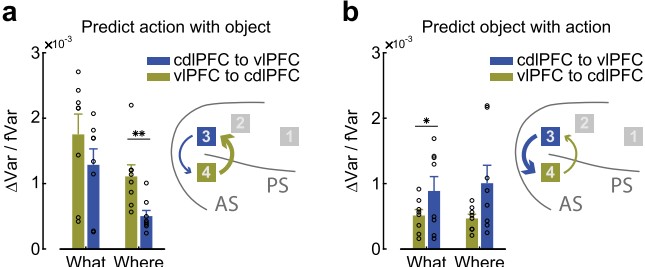

**Fig. 7 Information flow between ventral and dorsal–caudal LPFC.** Prediction variance between the cdlPFC and vlPFC. **a** Predicting action information with object information. **b** Predicting objects with action information. Inset: the location of each array and the flow direction and strength between them. Error bars represent mean ± SEM, n = 8 for each bar. A two-sided *t* test was used to compare two populations, *p = 0.0416, **p = 0.0069. Source data are provided as a Source Data file.

objects to actions. Consistent with this, we found that there was no difference in measures of information flow when compared between tasks in the initial trials of the block (Fig. 6e, f). However, as the animals determined the block type and the best choice, the difference emerged quickly (Movie S1). The difference was also lost during the reversal, and then it returned later in the block when the choice preference had switched (Fig. 6e, f).

To further examine the hypothesis that there is a ventro-dorsal gradient in representations in the caudal LPFC, we examined information flow between the cdlPFC and vlPFC in both task conditions (Fig. 7; Supplementary Fig. 10). We found that information flow was larger from the vlPFC to the cdlPFC when predicting action with an object (Fig. 7a; F (1, 28) = 5.90, p = 0.0219) and larger from the cdlPFC to the vlPFC when predicting object with action (Fig. 7b; F (1, 28) = 6.17, p = 0.019). Furthermore, the interaction between array direction (i.e., the vlPFC to the cdlPFC vs. the cdlPFC to the vlPFC) and cognitive process direction (i.e., object to action vs. action to object) was also significant (F (1, 57) = 12.15, p < 0.001). We also found that information flow from the vlPFC to the cdlPFC was stronger than the opposite direction when predicting action with an object in Where blocks (Fig. 7a, paired *t* test, t (7) = 3.78, p < 0.01). Information flow from the cdlPFC to the vlPFC was stronger than the opposite direction when predicting object with action in What blocks (Fig. 7b, paired *t* test, t (7) = 2.49, p = 0.042). Thus, when object locations define the chosen action, information about objects flows from the ventral to dorsal LPFC; when actions define the chosen object, information about actions flows from the dorsal to ventral LPFC.

We also carried out analyses on the LFPs recorded on each array, by computing cross-spectral power coupling between all pairs of electrodes (Supplemental Results). Cross-spectral coupling was strong between low frequencies (0–20 Hz, alpha, theta, and beta) in both the rostral-to-caudal (Supplementary Fig. 13a) and caudal-to-rostral (Supplementary Fig. 13b) directions. When we examined differences in the coupling (Supplementary Figs. 13c and 14), we found that coupling was stronger, particularly among beta frequencies in the rostral-to-caudal direction (Supplementary Fig. 13g–i), but stronger in the caudal-to-rostral direction, between alpha/theta and gamma frequencies (Supplementary Fig. 13f).

Taken together, these results show that information flow is stronger locally than distally. In addition, information flow from objects to actions is stronger from caudal to rostral and ventral to dorsal, and information flow from actions to objects is stronger from rostral to caudal and dorsal to ventral. Finally, flow from objects to actions was task-dependent, and developed with learning.

actions was stronger than information flow from actions to objects (compare Fig. 6c vs. d; F (1, 1517) = 45.6, p < 0.001).

We next examined information flow across learning. If the animals were using the object information to locate the object and generate a saccade, this should develop with learning and be disrupted at reversal. At the beginning of the block, when the monkeys were trying to determine the block type as well as the best choice, and during reversals, when the animals have to switch choice preference, there may be decreased information flow from

## Discussion

We examined the spatiotemporal representation and flow of information across the LPFC while monkeys carried out a two-armed bandit reinforcement learning task. When we examined the fraction of neurons encoding task variables, we found a caudo-rostral gradient, with a stronger and earlier representation of chosen actions and objects in the caudal relative to rostral LPFC. The vlPFC had a stronger representation of chosen objects than the cdlPFC, but the actions were represented similarly in these two areas. The reward was signaled simultaneously across the LPFC and most strongly in the cdlPFC. The results were generally consistent when we examined decoding instead of encoding. Interestingly, however, we found that there were more single cells significant for chosen objects than chosen actions across areas, whereas we were better able to decode chosen actions than chosen objects. This shows that although there were fewer neurons encoding actions, they did so more accurately[31]. It is also important to note that while we use the term "action" to refer to the saccade direction, the activity may represent visual–spatial processing and not motor planning per se, as we did not dissociate these factors. Previous work has examined this distinction within the region from which we recorded using an antisaccade task[32], and found that about 60% of neurons recorded across the area from which we recorded were visual–spatial, whereas 25% were action-related. Similar results have also been seen in delayed reach tasks[33].

Analysis of information flow supported and extended the analyses, which only looked at representations. We found that there was stronger information flow from caudal-to-rostral areas. This suggests that information about chosen actions and objects is first represented in caudal areas, after which it flows to rostral areas. The caudal–rostral gradient could have been driven by the use of eye movements, as the caudal arrays were near the frontal eye fields (FEF). Reaching movements may have led to increased activity in more rostral areas, given their connectivity with midline motor areas related to reaching movements[34]. We also found that task-relevant information flow across areas was specific to the required cognitive operation. Specifically, in What condition, there was increased information flow from objects to actions, when the monkeys had to use object information to direct a saccade. This information flow also tracked learning. There was stronger information flow from dorsal-to-ventral when predicting object with action information, and stronger information flow from ventral-to-dorsal when predicting action with object information. Analysis of LFPs showed that alpha/theta to gamma frequency power coupling was stronger from caudal to rostral, consistent with the single-neuron data. However, coupling among theta/alpha and beta showed a rostral-to-caudal flow. Some theories suggest that top-down and bottom-up information flow between areas in the sensory cortex utilizes different frequency channels[35,36]. However, whether similar ideas apply to flow within the LPFC is less clear.

While rodents have only a small region defined as the prefrontal cortex, primates have a large region that spans medial, orbital, and lateral prefrontal areas[37]. There is substantial anatomical and functional heterogeneity between these areas[38]. Even within the LPFC, there is considerable anatomical heterogeneity[39,40]. There are gradients of connectivity along both the ventro-dorsal and caudo-rostral axes[9]. The cdlPFC is more strongly connected to parietal areas important for spatial vision and oculomotor control, including the medial superior temporal and lateral intraparietal cortex, whereas the vlPFC is more strongly connected to temporal lobe visual areas[41,42]. In addition, the rdlPFC is connected to the medial parietal areas, including regions of the retro-splenial cortex[43]. However, there is also local connectivity within areas of the LPFC[44], which likely leads to local information flow. Based on

anatomical and functional considerations, proposals have been put forward, suggesting organization along both the caudo-rostral and ventro-dorsal axes of LPFC.

**Ventro-dorsal specialization in the caudal LPFC.** Several studies have suggested that there is a domain-specific organization along the ventro-dorsal axis of LPFC. Physiological recordings from monkeys trained to perform delayed-response tasks have suggested that the LPFC can be segregated into object and spatial domains. Neurons that code visual–spatial information are located in the cdlPFC, while those that code object identity information are located in the vlPFC[7,45,46]. These results in the visual domain have also been extended to auditory[47] and somatosensory[48] information. This proposal has been further supported by anatomical studies, which have shown that the dorsal regions of LPFC receive inputs from dorsally situated areas in the parietal visual or dorsal auditory cortex, whereas the ventral regions of LPFC receive inputs from the temporal lobe and ventrally situated auditory areas[41,49–51]. These findings suggest that the LPFC contains processing mechanisms for remembering what and where an object is[9], similar to what is found in the temporal parietal cortex[52]. Although there is separation across these processing streams, there is also substantial interaction and mixing[53–55].

In contrast to the "domain-specific" model, others have suggested that object and spatial information are integrated within the LPFC. To direct actions to appropriate objects, object identity and spatial location must be combined. The "integrative" model suggests that neuronal responses are shaped by the cognitive demands imposed by the task rather than the spatial location of the neurons. Miller and colleagues[11] employed a delayed-response task that required both memories of object identity and location. They found that some LPFC neurons showed only object-tuned (what) or location-tuned (where) delay activity. However, over half of the neurons with delay activity showed both what and where tuning. These neurons simultaneously reflected the location and identity of objects, and therefore they may play a role in integrating the identity and spatial location of objects in working memory[1].

Most previous electrophysiological data examining the role of LPFC in cognition were obtained using delayed-response tasks[7,11]. These tasks investigate the maintenance of action or object information over time. Our task was designed to address how the LPFC neurons dissociate the action and object information during rapid learning from reinforcement. We focused our analyses on the anatomical information flow between domains. We found data consistent with both "domain-specific" and "integrative" models. First, there was a stronger encoding of objects in the ventral than dorsal LPFC. However, there was only a trend toward an enhanced representation of actions in the dorsal relative to the ventral LPFC. Although many neurons tended to code both chosen objects and actions, more neurons tended to code only one domain (Supplementary Fig. 5), across both task conditions, as opposed to both domains within a task condition. When we predicted action with object information, we found enhanced information flow from the ventral to dorsal LPFC. When we predicted objects with action information, we found enhanced information flow from the dorsal-to-ventral LPFC (Fig. 7). Both effects were consistent across task conditions. These results suggest that there is anatomical segregation of information flow into the LPFC, followed by a rapid flow of information within the LPFC. Previous work suggested that local connectivity may account for the overlapping representation of spatial and object information in both the dorsal and ventral LPFC populations[11], which our analyses support.

**Caudo-rostral gradient in the LPFC**. In parallel with the ventro-dorsal organization of the LPFC, other groups have suggested a caudo-rostral organization (although spanning a larger expanse of the LPFC than we sampled). Supporting this hypothesis, there are differences in several anatomical features of the frontal cortex along this axis, including larger soma[56], reduced cell density[57], diminished intra-areal connectivity[58], more dendritic spines[59], lower myelination[57], decreased laminar differentiation[60], and longer connectional and synaptic distance from sensory input regions[25] in more anterior areas. Several groups have put forward models for the functional organization of the LPFC along the rostro-caudal axis[16,61]. For example, Badre and D'Esposito[18] manipulated the level of abstraction of stimulus-response rules required to make a choice and examined differences along this axis as a function of abstraction. Each level of abstraction increased the contingencies required to make a response. They found that activation in the more rostral LPFC regions tracked competition at higher abstraction levels, where abstraction was related to the number of factors that had to be integrated to respond correctly. By applying the same task, the authors found that frontal damage due to stroke impaired action decisions at a level of abstraction that was dependent on lesion location. Rostral lesions affected more abstract conditions and caudal lesions affected more concrete conditions[21]. Similar abstraction gradients have been observed in other human neuroimaging and lesion studies[19,22,61,62]. Collectively, these studies support the hypothesis that control at increased levels of abstraction requires areas located more rostrally in the frontal cortex.

Single-unit recording studies in macaques further support the hypothesis that there is a caudo-rostral organization in the LPFC. Riley et al.[23] found gradients of several aspects of information processing along this axis, including coding strength, response latency, and receptive field size, when they examined activity in untrained animals. Our results are consistent with it in the context of task-related activity. Along the rostro-caudal axis, more caudal neuronal populations showed stronger encoding (Figs. 2g and 3g) and co-occurrence of encoding (Supplementary Fig. 5g) for chosen action and object. This was also affected by the task context and learning process, with stronger coding of actions in Where blocks and stronger coding of objects in What blocks in the hold period in caudal areas (Figs. 2c and 3c). Our results also showed that the rdlPFC exhibited longer response latencies for processing object and action information, in agreement with prior studies[23].

Most studies supporting a rostro-caudal organization of the LPFC have suggested that the rdlPFC processes more abstract rules[16] or carries out domain-general feature integration operations[63]. The caudal LPFC, which is often taken to be the premotor cortex in human imaging experiments, on the other hand, is thought to carry out concrete operations[16]. However, other studies have shown that the rdlPFC regions can also be recruited by concrete operations like action selection[64] and the temporal, rather than the spatial activation profile of specific LPFC regions is modulated by maintenance demands, irrespective of the level of abstraction[65]. Tracer studies in monkeys have further shown that the structural network in the LPFC does not follow a strict rostro-caudal organization[27]. In our study, we found that caudal neuronal populations showed stronger responses and shorter response latencies to both action and object identity. Furthermore, when we examined the caudal–rostral representation of block type (i.e., What or Where), which is an abstract rule that defines the relevant learning dimension, we did not find an enriched representation more rostrally. We did not find any factors that dominated in the rdlPFC, although reward showed less of a gradient along this axis, with no difference in response latency. It is possible that if we had used a task with a different form of abstraction, we would have engaged the rdlPFC more strongly. For example, neurons that evaluate self-performance have been found in the rhesus monkey frontal pole, consistent with higher-order, metacognitive abilities residing in more rostral locations[66].

**Cortical information flow in the LPFC**. During cognitive processing, sensory information flows from early visual areas to parietal and temporal areas, and onto the prefrontal cortex. Choice signals develop simultaneously in frontoparietal regions and travel to the FEF and sensory cortex[67]. Several studies have examined the relative timing and strength of signals across connected cortical areas. However, this can only provide indirect evidence of how information flows[68–70]. Other studies have used approaches similar to ours and identified a specific neural signal related to the executive control of cognition that is transmitted across cortical areas[71]. The authors simultaneously recorded the activity of neurons in the LPFC and posterior parietal cortex (PPC) of monkeys performing a rule-based spatial categorization task. They used a decoding analysis to "read out" the category, and then computed the correlation between whitened time series in the two areas at different time lags. The results showed that the decoded time series in the LPFC was correlated with the time series in the PPC at positive lags, which suggested that categorization signals were transmitted asymmetrically in a top-down direction from the LPFC to the PPC. A similar method has been used in an object construction task in which the authors found that retina-centered visual information could be used to predict subsequent object-centered signals, but not vice versa[72] when monkeys were required to map from retina-centered to object-centered coordinates to carry out a task.

Since our task included What and Where blocks, the monkeys needed to use either action or object information to make a choice in each trial. By adopting a causal analysis framework, similar to the method used in a previous study[71], we measured how task-relevant neural signals were transmitted across subregions in the LPFC. Our results showed that information flow in the caudo-rostral direction was stronger than in the rostro-caudal direction when processing the action but not the object information. This was consistent with our decoding analyses, in which we found that the caudal neuronal populations had stronger (Fig. 3g) and faster representations of action (Fig. 3h). Since the block type was not indicated by any explicit cues in our task, the monkeys needed to use both action and object information to guide their behavior, especially at the beginning of each block. To investigate how this happened, the information flow from actions to objects and from objects to actions within the LPFC was calculated. We found that information flow from objects to actions was stronger in the caudo-rostral direction, especially in What blocks. The task-dependent effect of flow from objects to actions also developed with learning (Fig. 6f). Although it was weaker, information flow from action to object also showed a rostral-to-caudal gradient. This might be due to the stronger encoding of action than object information in the rdlPFC (Fig. 3a, b, d, e).

In conclusion, we found a substantial caudo-rostral gradient in the strength and response latencies of information relevant to both variables. We also found a caudo-rostral flow of information. When we specifically compared the dorsal and ventral areas in caudal LPFC, we found an enhanced representation of chosen objects in the vlPFC. We also found that there was more information flow from chosen objects to chosen actions in the ventral-to-dorsal direction, and more flow from chosen actions to chosen objects in the dorsal-to-ventral direction. Therefore, our analyses support a model in which information about chosen

objects first flows into the vlPFC, and information about chosen actions first flows into the cdlPFC. Following this, there is flow within the LPFC, and from the caudal-to-rostral LPFC. Thus, our analyses support both anatomical segregation and rapid physiological integration of information relevant to reward-related choices within the LPFC.

## Methods

**Subjects.** Two male monkeys (*Macaca mulatta*, W—6.7 kg, age 4.5 yo, V—7.3 kg, age 5 yo) were used as subjects in this study. For the duration of the study, the monkeys were placed on water control and earned their fluid through their performance on the task on testing days. Experimental procedures for all monkeys were performed following *the Guide for the Care and Use of Laboratory Animals* and were approved by the National Institute of Mental Health Animal Care and Use Committee.

**Experimental setup.** Monkeys were trained to perform a saccade-based two-armed bandit task for juice rewards[28]. Stimuli were presented on a 19-inch liquid crystal display monitor situated 40 cm from the monkey's eyes. During training and testing, monkeys sat in a primate chair with their heads restrained. Stimulus presentation and behavioral monitoring were controlled by a PC running Monkeylogic (version 1.0), a MATLAB-based behavioral control program[73]. Eye movements were monitored at 400 fps using an Arrington Viewpoint eye tracker (Arrington Research, Scottsdale, AZ) and sampled at 1 kHz. On rewarded trials, a fixed amount of undiluted apple juice (0.08–0.17 ml) was delivered through a pressurized plastic tube gated by a computer-controlled solenoid valve[74].

**Task design and stimuli.** The monkeys were trained to complete around 24 blocks per session (Fig. 1a, b). The task has been described in detail previously[28,75]. Each block consisted of 80 trials and one reversal of the object-based or action-based reward contingencies. On each trial, monkeys had to acquire and hold a central fixation point for a random interval (400–600 ms). After the monkeys acquired and held central fixation, two objects appeared one each to the left and right (6° visual angle from fixation) of the central fixation point. The monkeys reported their choices by making a saccade to their selection, which could be based on the object or the direction of their saccade. After holding their choice for 400 ms, a reward was stochastically delivered according to a 70%/30% reward schedule. If the monkeys failed to acquire central fixation within 5 s, hold central fixation for the required time, or make a choice within 1 s, the trial was aborted and then repeated.

Each block used two novel objects that were randomly assigned to the left or right side of the fixation point for every trial. The objects were changed across blocks but remained constant within a block. What and Where blocks were randomly interleaved throughout the session, and block type was not indicated to the monkey. For What blocks, reward probabilities were assigned to each object independently of the saccade direction to select an object. Conversely, for Where blocks, reward probabilities were assigned to each saccade direction independently of the objects presented on either side of central fixation. The block type (What or Where) was held constant for each 80-trial block. One of the objects or one of the actions had a lower probability (30%) of being rewarded, and the other had a higher probability (70%). The trial in which the reward mapping reversed in each block was randomly selected from a uniform distribution from trial 30 to 50, inclusive. The reversal trial was independent of the monkey's performance and was not signaled to the monkey[75].

**Data acquisition and preprocessing.** Microelectrode arrays (Blackrock Microsystems, Salt Lake City, USA) were surgically implanted over the LPFC, surrounding the principal sulcus (Fig. 1d). Four 96-electrode (10 × 10 layouts) arrays were implanted in each hemisphere. Details of the surgery, implant design[76] and data acquisition[29,75] have been described previously. Briefly, a single bone flap was temporarily removed from the skull to expose the LPFC. Then the dura mater was cut open to implant the electrode arrays into the cortical parenchyma. The dura mater was then sutured, and the bone flap sewn back into place with absorbable sutures to protect the brain and the implanted arrays. Meanwhile, a 3D-printed biocompatible connector holder was implanted onto the posterior portion of the skull. Neurophysiology recording for all monkeys began after they had recovered from the implant surgery.

Recordings were made using the Grapevine System (Ripple, Salt Lake City, USA). Two neural interface processors (NIPs) made up the recording system, one NIP (384 channels) was connected to the four multielectrode arrays of each hemisphere. Behavioral codes from MonkeyLogic and eye-tracking signals were split and sent to each Ripple box. The raw extracellular signal was high-pass filtered (1-kHz cutoff) and digitized (30 kHz) to acquire the single-unit activity. Spikes were detected online, and the waveforms were stored using the Trellis package (Grapevine). Single units were manually sorted offline. The threshold for spike acquisition was set at 4.5 × root to the mean square of the baseline signal for each electrode.

**Neural data.** We collected data in eight recording sessions (four sessions per animal). To identify task-related neurons, all trials on which monkeys chose one of the two stimuli were analyzed. Trials in which the monkey broke fixation and failed to make a choice were excluded. On valid trials, the firing rate of each cell was computed in 50-ms bins, advanced in 10-ms increments, and time-locked to the cue onset. We fit a sliding window fixed-effect ANOVA to these windowed spike counts. The ANOVA included factors for the chosen object, chosen action, reward, and value. The value factor served to model value updating[77]. All other factors were modeled as nominal variables. Significant encoding for each time bin and factor was evaluated at $p < 0.05$.

We fit Rescorla–Wagner reinforcement learning models to the choice data for each block type. Models were fit with separate learning rates and inverse temperatures for the two-block types. In the mode, value updates were given by:

$$v_i(k + 1) = v_i(k) + \delta_f(R - v_i(k)) \tag{1}$$

where $v_i$ is the value estimate for option $i$, $R$ is the outcome for the choice for trial $k$, and $\delta_f$ is the outcome-dependent learning rate parameter, where $f$ indexes whether the current choice was rewarded ($R = 1$) or not ($R = 0$), i.e., $\delta_{pos}, \delta_{neg}$. For each trial, $\delta_f$ is one of two fitted values used to scale prediction errors based on the type of reward feedback for the current choice. We then passed these value estimates through a logistic function to generate choice probability estimates:

$$d_1(k) = \left(1 + e^{\beta(v_2(k) - v_1(k))}\right)^{-1}, \, d_2(k) = 1 - d_1(k) \tag{2}$$

The likelihood for these models is given by

$$f\left(x, y | \beta, \delta_{pos}, \delta_{neg}\right) = \prod_k [d_1(k)c_1(k) + d_2(k)c_2(k)] \tag{3}$$

where $c_1(k)$ had a value of 1 if option 1 was chosen on trial $k$ and $c_2(k)$ had a value of 1 if option 2 was chosen. Conversely, $c_1(k)$ had a value of 0 if option 2 was chosen, and $c_2(k)$ had a value of 0 if option 1 was chosen for trial $k$. We used standard function optimization methods to maximize the likelihood of the data given the parameters.

A four-way ANOVA was applied to examine the encoding of task variables after cue onset and during the hold period. It was a standard, non-nested, linear model with two levels of interaction. The factors included cerebral hemisphere (i.e., left and right hemisphere), block type (i.e., What or Where blocks), domain type (i.e., action or object identity), and array locations (i.e., from array 1 to array 4).

To detect the response latencies, a paired $t$ test was performed between the average percentage of task-related neurons of the baseline period (from −1.5 to −0.5 s from cue onset) and each bin across the whole trial time course from all eight arrays. The first time point that showed a significant difference was defined as the response latency. A four-way ANOVA was applied to examine the difference of response latencies from each region, which was carried out with leave-one-session-out. The ANOVA was a standard, non-nested, linear model and with two levels of interactions. The factors included cerebral hemisphere, block type, domain type, and array locations.

**Decoding analyses.** We carried out the decoding analysis on chosen actions, chosen objects, and rewards (i.e., reward or nonreward). Therefore, for these analyses, the chance performance was 50%. Analyses were carried out using leave-one-trial-out cross-validation. The model was fit with the remaining trials and tested on the trial that was held out of the analysis. All simultaneously recorded neurons from each array were used to predict the indicated factor. Decoding was computed in 20-ms bins, advanced in 20-ms increments, and time-locked to cue onset. Neural activity was not normalized or transformed. Raw spike counts in 20-ms bins for each neuron were used.

The posterior probability of choice, which is the probability of selecting the more rewarding action or object over trials, was calculated by

$$p_i(t) = \frac{\exp\left(-\left(x_k(t) - \overline{X_i}(t)\right)^2\right)}{\sum_{j=1:2} \exp\left(-\left(x_k(t) - \overline{X_j}(t)\right)^2\right)} \tag{4}$$

Here, $p_i(t)$ represents the choice probability for option $i$ at time t, the vector $\mathbf{x_k(t)}$ represents the neural population activity, with each element of the vector representing the spike count of a single neuron, in a single trial $k$, at time $t$. The vector $\overline{X_i}(\mathbf{t})$ represents the mean neural population activity across trials for a chosen object or action that was indicated by $i$ or $j$. This is a linear decoder which, in a probabilistic sense, would be a linear Gaussian decoder with a spherical covariance matrix.

**Information flow analyses.** A regression model was used to measure the flow of information across the subregions of the LPFC

$$p_i(\text{choice} | r_i(t)) = a_0 + \sum_{j=1:10} a_{(i,j)} p_i(\text{choice} | r_i(t - j)) \\ + \sum_{k=1:8 \setminus i} \sum_{l=0:10} a_{(k,l)} p_k(\text{choice} | r_k(t - l)) \tag{5}$$

Here, $p$ represents the posterior probability of choice, $a$ represents the kernel coefficients related to the input arrays, $i$ represents the output array (one of the eight arrays), $j$ represents the lagged bin number ahead of time $t$ in the output array, $k$ represents the input arrays, and $l$ represents the bin number ahead of time $t$ in the output array. Choice probability was computed from $-0.5$ to $1.5$ s from cue onset, in 20-ms bins, advanced in 20-ms increments. Note that we are predicting the future information, $p_i(\text{choice}|r_i(t))$, using past information on the same array, $\sum_{j=1:10} a_{(i,j)} p_i(\text{choice}|r_i(t-j))$ and current and past information on other arrays, $\sum_{k=1:8\backslash i} \sum_{l=0:10} a_{(k,l)} p_k(\text{choice}|r_k(t-l))$. When we tested for the effect of one array on another, we dropped the array under consideration from the sum, $\sum_{k=1:8\backslash i} \sum_{l=0:10} a_{(k,l)} p_k(\text{choice}|r_k(t-l))$ and compared the prediction of $p_i(\text{choice}|r_i(t))$ with and without the array under consideration.

**Reporting summary**. Further information on research design is available in the Nature Research Reporting Summary linked to this article.

## Data availability

The data that support the findings of this study are available from https://data.mendeley.com/datasets/m4f38w49fb/1. Citation: Tang, Hua; Bartolo, Ramon; Averbeck, Bruno (2020), "Dataset for studying information flow among macaque lateral prefrontal cortex", Mendeley Data, V1, https://doi.org/10.17632/m4f38w49fb.1. Source data are provided with this paper.

## Code availability

Custom codes are available on GitHub (https://github.com/CHT2016/information_flow).

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

## Acknowledgements

We thank Christos Constantinidis for valuable comments. This work was supported by the Intramural Research Program of the National Institute of Mental Health (ZIA MH002928).

## Author contributions

R.B.O. and B.A. designed the research; R.B.O. and B.A. performed the research; H.T. and B.A. analyzed the data; H.T. and B.A. wrote the paper.

## Competing interests

The authors declare no competing interests.
