## [Peer Review File · Nature Communications]

REVIEWER COMMENTS

Reviewer #1 (Remarks to the Author):

Tang et al. have taken on the admirable goal of analyzing physiological gradients across the primate prefrontal cortex in the caudal-rostral and dorsal to lateral domains. Previous anatomical, physiological and imaging studies have implicated the existence of such a topography, and the Averbeck lab has the physiological and computational resources to tackle this intriguing and important question. The current paper manages to capture some interesting data in support of these gradients, but does so using a suboptimal experimental design that lessens enthusiasm for the study overall. This is particularly worrisome because of the confound between action and place/space, which limits and confuses interpretations of the data.

There are likely multiple, interacting gradients across the primate PFC. For example, anatomical, physiological and imaging data indicate a dorsal to ventral gradient for sensory inputs, where spatial information is brought to more dorsal locations, while sensory features are represented more ventrally. There also may be a gradient in motor response, e.g. with the caudal principal sulcus interacting with the nearby frontal eye fields for saccadic responses, and more rostral portions of principal sulcus connecting with the cingulate motor areas involved with arm-hand motor responses (e.g. Bates and Goldman-Rakic. *J Comp Neurol.* 1993;336(2):211-228). (This gradient would help to explain why the authors saw so little gradient for action, and why the strongest signals were caudal, since all of the motor responses in the current study were eye movements.) There may be additional caudal to rostral gradients in cognitive operations, e.g. with simpler operations more caudal, and more abstract operations farther rostral (consistent with the metacognitive properties even seen in monkeys, e.g. Tsujimoto et al, *Nat Neurosci.* 3: 120-6, 2010). This was mentioned in the discussion, but was not a dimension explicitly manipulated in the current study.

The current experimental design seems to be a lost opportunity- ideally they would have a task that explicitly manipulated all of these dimensions, without the confounds between motor and sensory dimensions that occurred with the current design. The authors are aware of this problem, writing: "It is also ^{SEP}important to note that while we use the term "action" to refer to the saccade direction, the activity may represent visual-spatial processing and not motor planning per se." Thus, it is disappointing that they did not design the study properly to begin with.

Small points-

Figure 1D should label areas rdIPFC, mdIPFC, cdIPFC and vlIPFC to correspond with the remaining figures.

The authors should cite the more pertinent Ungerlieder human imaging reference: Ungerleider LG1, Courtney SM, Haxby JV. A neural system for human visual working memory. *Proc Natl Acad Sci U S A.* 1998 Feb 3;95(3):883-90.

Reviewer #2 (Remarks to the Author):

In the manuscript "Meso-scale functional organization of macaque lateral prefrontal cortex" Tang et al., test in two monkeys and multiple 100-electrode arrays implanted in lateral prefrontal cortex in two hemispheres how neuronal information about chosen objects and chosen locations vary as a function of the electrode array location. The authors use a probabilistic learning task where blocks of trials had the same reward rule and pairs of successive blocks the same target dimension (location or object identity) in four experimental recording sessions, each with behavioral and neural data from ~24 learning blocks.

The study analyzes in more than 6000 neurons how much information about the chosen object and the chosen action-direction is encoded by single cells and decodable from the population of cells. It then quantifies the information about the choice of the monkeys given neural activity (indexed as the posterior probability from the decoding model) and assessing how the information at different recording sites is related to each other using (auto)regressive (Granger) modeling. To quantify the information flow from a specific array to other arrays they calculate how that array contributes to the overall (full) information when considering all arrays versus all-but-that array. With that approach the relative direction of (the predictability of) information is quantified.

The results from this analysis approach confirm many prior insights which for this paper is a strength given the novelty of the learning task and the information analysis results. These insights include that relevant visual objects are encoded in ventrolateral PFC, that relevant locations are encoded more dorsally, and that caudal PFC parts have more task informative neurons. Beyond this basic representational organization the study finds that the direction of information about chosen objects/locations show systematic anatomical gradients as well.

The study then reports that object to action information can be predicted with a gradient from caudal to rostral in What blocks, which develops with learning; while within the caudal PFC the gradient was such that object information in ventrolateral PFC predicted action information in dorsal PFC (Fig 6 and 7).

The study is exceptional in scope (spatial extent and density of recordings) which allowed describing not only the well known object / location ventral / dorsal gradient, but to quantify the relative "direction" of information transmission about chosen objects/locations along the caudo-rostral and rostrocaudal axis. These later results are innovative and novel as they suggest that segregated encoding of chosen objects/locations translate into an integration of functional encoding with a fast time course within a trial (shortly after choosing an object) _and_ with a systematic time course over trials during learning the relevant block rule and rewarded object/location.

There are, however, various aspects of the paper that deserve clarification.

- It is not clear why the spatial specific prediction of post. probabilities is considered to reflect information flow. The time lag is not analyzed in detail so the "flow" component is not explicitly statistically tested. It seems that the analysis result identifies a significant effect when the decoding in one area is similar to the decoding in another area, and the directionality comes into play when the decoding in one area shows higher accuracy in one area than in the other predicting areas (or 'arrays' instead of areas). If the effect is based on similarity and strength of decoding in one over another area/array, it is not justified to consider this as reflecting information flow. Since the applied method is novel and terminology can be misleading for novel/not established techniques it seems important that the the paper includes a more explicit definition and consideration of the approach.
- Related to the previous aspect, the directional information analysis includes time lag parameters, but the presentation of the results do not allow discerning a possible time lag of predictability of information in one array from the information that built up in the other arrays (partial model analysis and its difference to the full model). To reflect a directionality of prediction it is necessary to quantify and visualize how much information is there in the sender before it is available in the receiver.
- It is difficult to understand the effect size of the 'information' / posterior probability reflects. Given the decoding accuracy of 60-80% how much _more_ information is conveyed by caudolateral choice related activity. Can the authors translate their measures into a more common measure (to learn how many choices are better predicted from caudolateral than from rostralateral arrays)? Addressing this aspect might also help to understand why the author interpret the found gradients as

"substantial" (line 521).

- The study assumes and conveys explicitly that neurons in prefrontal cortex encode the identity of an object or a spatial location, which is misleading. The analysis operationalizes the encoding of an object (or location) that is chosen (and fixated for 400ms). What is encoded is then the 'chosen object identity' and the 'chosen location'. This is different to conveying that an object identity is encoded. I am not sure whether the authors can analyze object identity as they always show the same two objects in a block and hence only the choice and the location of these objects vary. The task design does not allow distinguishing whether a neuron encodes within a block the rewarded object or the nonrewarded object (both are present on the screen), only whether the chosen object is differently encoded than the nonchosen object. This should be clarified to prevent readers to wrongly understand that objects or action are encoded independent of choices in these areas.
- The abstract states that the study "examined information timing and flow across". ... but the latencies of the information flow is not statically analyzed in depth.
- The abstract lacks focus and is very general. Given the specific findings of a gradient can the abstract not introduce more explicitly the concepts of segregation (by domains) and integration (by functional requirement) and highlight the novel aspects of the results including also the fact that the information analysis allows to track learning the relevant objects ?
- It is not clear whether the main gradient results were evident similarly in each of the two monkeys, and in each hemisphere. How reproducible are these results ?
- Fig 6: the legend and axis labels are very difficult to understand. Please describe the axis meaning more explicitly in the legend.
- A key paper showing dorsal-ventral gradient of space/object coding in PFC that is not yet considered but of particular importance is:
Lebedev MA, Messinger A, Kralik JD, Wise SP (2004) Representation of attended versus remembered locations in prefrontal cortex. PLoS Biol 2: e365.
This reviewer has no ties to these authors.
- The methods section about the decoding analysis does not describe what the input was for the decoder (single neuron data), how the neural data was normalized or not prior to decoding, what the decoding method actually was and how the data from the different monkeys were combined.
- Because neural response latencies are analyzed with differences in what and where blocks it seems important to know whether the initial behavioral response latencies in both blocks types were similar (were behavioral latencies also slower in the action blocks?).
- How did the authors use neurons recorded from the same array at successive days? Is it important to distinguish whether neurons on successive days are identical and thus should not be counted and analyzed twice?
- It would help to learn about the sorting criterion used to characterize action potentials to be from a single neuron. If the action potentials are from different neurons or are not separated from hash it is important to convey that the data are based on multiunit activities. Were effects dependent on the isolation quality of the neurons. This would simplify future studies aimed to reproduce the effects based on multiunit activity.
In this context it is unclear what the average firing rate of the neurons were.
- The methods section misses a description of the information flow analysis was statistically tested.

- The methods section has no information about the cross-spectral analysis. Which method, time window, statistics, etc. was used?
- The study used arrays from both hemispheres but it is difficult to discern whether the main effects of the study were similarly evident in each hemisphere or whether the gradients and the encoding/decoding showed some hemisphere specificity.
- In the results section it does not become clear why the decoding analysis was done after already quantifying encoding? An additional sentence motivating the analysis seems useful.

Reviewer #3 (Remarks to the Author):

Review – Meso-scale functional organization of macaque lateral prefrontal cortex
Tang, Bartolo, and Averbeck

The manuscript by Tang and colleagues describes the findings of an heroic set of experiments in which the authors set out to address a question of long-standing interest with respect to the organization of the lateral prefrontal cortex – whether the representation of different types of task-relevant information are represented preferentially in specific LPFC subregions. To address this question, 4 utah arrays were implanted bilaterally in two monkeys, each covering a specific LPFC subregion (rdIPFC, mdIPFC, cdIPFC, and vdIPFC). Monkeys performed a well-established task – a two-armed bandit task in which saccades were rewarded stochastically based on either the identity of a target object or the location to which the saccade was directed in a series of separate trial blocks. Population analyses of encoding, decoding, and information flow across arrays were conducted to determine whether representations of objects or actions were more prevalent in any PFC subregion, whether the latencies of these representations varied between regions, and how information regarding objects and actions flowed between different subregions. The authors conclude that:

- 1) more neurons are responsive during the task in caudal than rostral PFC
- 2) Both chosen objects and chosen actions are encoded in vdIPFC and cdIPFC, though a bias exists toward chosen objects in vdIPFC and chosen actions in cdIPFC
- 3) Response latencies of the population followed a caudal-rostral gradient, with latencies being shorter in caudal LPFC
- 4) Response latencies for reward information were similar across arrays.
- 5) Co-occurrence of action and object representations was stronger in caudal than rostral LPFC
- 6) Decoding performance for predicting chosen object or actions was greater for caudal than rostral LPFC. Similar caudal-rostral gradient of response latencies as encoding analysis.
- 7) Information flow was strongest between neighbouring arrays. Overall information flow was strongest in caudo-rostral direction when predicting actions.
- 8) Depending on task requirements, information regarding actions flows from dorsal to ventral LPFC, and information about objects flows from ventral to dorsal LPFC.

Overall, the authors conclude that several gradients exist in the caudo-rostral direction. Information carried by single neurons, as was as information flow is strongest in this direction. This is also the case for response latency, and information about both object identity and location are both present. A

bias toward representing action information is present in cdIPFC, and a bias toward object information is present in vdIPFC. Information about action is transmitted from cdIPFC to vdIPFC when actions are relevant, and from vdIPFC to cdIPFC when object identity is relevant to the task at hand.

This is a thoughtfully designed and skillfully executed series of experiments. The experimental task is appropriate to address the questions at hand, the quality of the data is high, and all the analyses reported are sound. The conclusions flow correctly from the analyses as described. Although in one sense the reporting of mixed action and object selectivity across PFC subregions is expected from the large body of literature in this field, the authors have been able to go a step farther and directly address questions of information flow across PFC by recording from large samples of neurons in specific PFC subregions simultaneously. This represents a substantial advance and allows the novel conclusions generated by their analyses of information flow. I would recommend publication of this study. I have a few questions and comments, outlined below.

1) If I understand correctly, in their latency analyses, the authors report the latency at which, during the task, the percentage of task-related neurons exceeds that observed in the baseline period. Essentially my understanding here is that this represents the latency at which the population of neurons carrying task-relevant representations of object or action come "on-line" during the task. I wonder if a further analysis at the single-neuron level might shed more light on the differences between subregions here. Could the authors conduct an analysis to determine the latency at which single neurons discriminate the preferred from non-preferred object during the 'what' blocks, and the preferred from non-preferred action during the 'where' blocks? A sliding ROC or the sliding t-test type of analysis could be used to compare these two activity profiles and a discrimination time could be determined for each single unit and averaged across the different arrays to compare latency differences. Such an analysis might complement the existing population-level analysis done here.

2) The authors report that the reaction times for the 'where' trials was significantly shorter than that for 'what' trials. However, in the latency analysis outlined, it is demonstrated that the latency for object representations is significantly shorter than that for actions. I wonder then at the behavioural relevance of the signals reported here. I wonder if a more direct analysis of the relationship between activity in the different PFC subregions and reaction times might add something here. For example, could the authors investigate the trial-by-trial correlation between activity at the end of the fixation period and the reaction times of the forthcoming saccades? Although, as the authors correctly point out, these signals are not explicitly "motor", we know that they are most likely being sent to areas that are in order to bias behavior appropriately. Also, given the seeming increasing abstraction of information represented along the caudo-rostral gradient, it might be expected that significant correlations would be obtained for the caudal but not rostral LPFC areas, thus showing further differences between regions.

3) Much, if not most of the electrophysiological evidence investigating domain-specificity in macaque PFC has been obtained by investigating the nature of persistent activity during delay periods of oculomotor delayed-response or delayed-match-to-sample tasks. Here the authors have used a different type of task and are investigating differences in activity during task epochs involving fixation before and after onset of choice stimuli in different conditions. I don't see this as any major issue, however, some acknowledgement of and comment on this would be appropriate in the discussion.

We are grateful to the reviewers for their careful reading of our manuscript, their insightful comments, and their overall positive evaluation. We have made extensive changes to address all issues raised by the referees (*in italics below*).

Reviewer #1 (Remarks to the Author):

Tang et al. have taken on the admirable goal of analyzing physiological gradients across the primate prefrontal cortex in the caudal-rostral and dorsal to lateral domains. Previous anatomical, physiological and imaging studies have implicated the existence of such a topography, and the Averbeck lab has the physiological and computational resources to tackle this intriguing and important question. The current paper manages to capture some interesting data in support of these gradients, but does so using a suboptimal experimental design that lessens enthusiasm for the study overall. This is particularly worrisome because of the confound between action and place/space, which limits and confuses interpretations of the data.

Comment 1a: There are likely multiple, interacting gradients across the primate PFC. For example, anatomical, physiological and imaging data indicate a dorsal to ventral gradient for sensory inputs, where spatial information is brought to more dorsal locations, while sensory features are represented more ventrally. There also may be a gradient in motor response, e.g. with the caudal principal sulcus interacting with the nearby frontal eye fields for saccadic responses, and more rostral portions of principal sulcus connecting with the cingulate motor areas involved with arm-hand motor responses (e.g. Bates and Goldman-Rakic. J Comp Neurol. 1993;336(2):211-228). (This gradient would help to explain why the authors saw so little gradient for action, and why the strongest signals were caudal, since all of the motor responses in the current study were eye movements.)

Response: We thank the reviewer for this comment. We agree that there are likely multiple gradients within PFC, which is itself very complex and heterogeneous. One may want to build a task that had more factors crossed to address the multiple models that have been put forward. We have been focused on the spatial vs. object gradient in the context of learning, and we have been studying this across various frontal-striatal circuits, which is why we were interested in examining this distinction in IPFC. As the reviewer points out, this distinction has been put forward in previous work. However, it has not been examined in the context of learning with simultaneously recorded activity. Therefore, we felt that our study was a useful contribution.

We also agree that there are important distinctions between reaches and eye movements. And we may pursue this in future studies. We have added the following to the discussion to address this, **"The caudal-rostral gradient could have been driven by the use of eye movements, as the caudal arrays were near the frontal eye fields. Reaching movements may have led to increased activity in more rostral areas, given their connectivity with midline motor areas related to reaching movements³⁴."**

Comment 1b: There may be additional caudal to rostral gradients in cognitive operations, e.g. with simpler operations more caudal, and more abstract operations farther rostral (consistent with the metacognitive properties even seen in monkeys, e.g. Tsujimoto et al, Nat Neurosci. 3: 120-6, 2010). This was mentioned in the discussion, but was not a dimension explicitly manipulated in the current study.

The current experimental design seems to be a lost opportunity- ideally they would have a task that explicitly manipulated all of these dimensions,...

Response: With respect to additional gradients, our task has an important abstraction, which is the block type. This is a hierarchical rule that defines which domain is currently relevant. In our

task it is inferred, rather than instructed, and therefore it is more akin to rule discovery in the Wisconsin Card Sorting task. We now include analysis of activity related to the block type as a supplemental figure for completeness and we refer to it in the discussion (see below). Therefore, we do think our task taps into several factors that have been put forward with respect to the anatomical organization of the prefrontal cortex. Although more could have been addressed, “We also examined the encoding of the blocktype at a single cell level (Supplementary Fig. 4). The blocktype is an abstract rule that defines the relevant choice dimension. When we examined blocktype we found that it was more strongly represented in caudal areas (Following cue period, Array; $F(3, 56) = 12.14, p < 0.001$), similar to the other factors. Therefore, we did not find an enhanced representation of blocktype in more anterior parts of LPFC.”

Fig. S4. Population encoding of chosen action and chosen object. Percentage of task related neurons in each region that encoded block types. Shaded zones represent mean ± SEM, $n = 8$ for each line. The black * symbols at the top indicate a significant difference among the four regions (1-way ANOVA, $p < 0.01$). The colored * symbols indicate a significant difference (paired t-test, $p < 0.01$) of task-related neuron percentage between the corresponding region and its baseline.

*Comment 1c: ...without the confounds between motor and sensory dimensions that occurred with the current design. The authors are aware of this problem, writing:
"It is also important to note that while we use the term "action" to refer to the saccade direction, the activity may represent visual-spatial processing and not motor planning per se."
Thus, it is disappointing that they did not design the study properly to begin with.*

Response: With respect to spatial coding vs. action coding, this is also an importation question, and we did not explicitly dissociate these dimensions in our study. This has been, however, investigated previously. To address this, we have further elaborated on this in the discussion: “It is also important to note that while we use the term “action” to refer to the saccade direction, the activity may represent visual-spatial processing and not motor planning per se, as we did not dissociate these factors. Previous work has examined this distinction within the region from which we recorded using an anti-saccade task ³², and found that about 60% of neurons recorded across the area from which we recorded were visual-spatial, whereas 25% were action related. Similar results have also been seen in delayed reach tasks ³³.”

Small points-

1. *Figure 1D should label areas rdIPFC, mdIPFC, cdIPFC and vlPFC to correspond with the remaining figures.*

Response: We have modified Fig. 1 and placed the labels on top of the arrays on the right-hand image.

2. *The authors should cite the more pertinent Ungerlieder human imaging reference:*

Ungerleider LG1, Courtney SM, Haxby JV. A neural system for human visual working memory. Proc Natl Acad Sci U S A. 1998 Feb 3;95(3):883-90.

Response: The suggested reference has been cited in the manuscript:

Support ^{7, 8, 9, 10} for this proposal derives from studies that show that spatial stimuli recruit the caudal dorsolateral prefrontal cortex (cdIPFC) and object or verbal stimuli recruit the ventrolateral prefrontal cortex (vlPFC).

Reviewer #2 (Remarks to the Author):

In the manuscript "Meso-scale functional organization of macaque lateral prefrontal cortex" Tang et al., test in two monkeys and multiple 100-electrode arrays implanted in lateral prefrontal cortex in two hemispheres how neuronal information about chosen objects and chosen locations vary as a function of the electrode array location. The authors use a probabilistic learning task where blocks of trials had the same reward rule and pairs of successive blocks the same target dimension (location or object identity) in four experimental recording sessions, each with behavioral and neural data from ~24 learning blocks.

The study analyzes in more than 6000 neurons how much information about the chosen object and the chosen action-direction is encoded by single cells and decodable from the population of cells. It then quantifies the information about the choice of the monkeys given neural activity (indexed as the posterior probability from the decoding model) and assessing how the information at different recording sites is related to each other using (auto)regressive (Granger) modeling. To quantify the information flow from a specific array to other arrays they calculate how that array contributes to the overall (full) information when considering all arrays versus all-but-that array. With that approach the relative direction of (the predictability of) information is quantified.

The results from this analysis approach confirm many prior insights which for this paper is a strength given the novelty of the learning task and the information analysis results. These insights include that relevant visual objects are encoded in ventrolateral PFC, that relevant locations are encoded more dorsally, and that caudal PFC parts have more task informative neurons. Beyond this basic representational organization the study finds that the direction of information about chosen objects/locations show systematic anatomical gradients as well.

The study then reports that object to action information can be predicted with a gradient from caudal to rostral in What blocks, which develops with learning; while within the caudal PFC the gradient was such that object information in ventrolateral PFC predicted action information in dorsal PFC (Fig 6 and 7).

The study is exceptional in scope (spatial extent and density of recordings) which allowed describing not only the well known object / location ventral / dorsal gradient, but to quantify the relative "direction" of information transmission about chosen objects/locations along the caudo-rostral and rostrocaudal axis. These later results are innovative and novel as they suggest that segregated encoding of chosen objects/locations translate into an integration of functional encoding with a fast time course within a trial (shortly after choosing an object) _and_ with a systematic time course over trials during learning the relevant block rule and rewarded object/location.

Response: We thank the reviewer for this positive evaluation.

There are, however, various aspects of the paper that deserve clarification.

1a. It is not clear why the spatial specific prediction of post. probabilities is considered to reflect information flow. The time lag is not analyzed in detail so the "flow" component is not explicitly statistically tested.

1b. It seems that the analysis result identifies a significant effect when the decoding in one area is similar to the decoding in another area, and the directionality comes into play when the decoding in one area shows higher accuracy in one area than in the other predicting areas (or 'arrays' instead of areas). If the effect is based on similarity and strength of decoding in one over another area/array, it is not justified to consider this as reflecting information flow.

Since the applied method is novel and terminology can be misleading for novel/not established techniques it seems important that the paper includes a more explicit definition and consideration of the approach.

Response: This is an important point and we apologize for not being clearer. We have made changes throughout the section which describes these results, which hopefully clarifies our approach. The analysis framework is similar to Granger causality. It is considered directed

information flow because we are predicting future activity in one array with past activity in another. We tested information flow statistically by comparing the amount of flow in each direction (i.e., caudal-rostral vs. rostral-caudal) and also across conditions. This was in Fig. 6. Because connectivity is recurrent, there will be detectable flow in each direction. And therefore, we tested whether it was stronger in one direction or another, rather than present/absent. Please see the updated text in that section for further details.

We have added an additional analysis to further support our assertion that this is flow (supplementary Fig. 12). In the original analysis, included in the paper, the variable l in our regression equation (see third summation symbol in regression equation below) takes on values 0-10 (Fig. R1A1 and R1B1). Thus, we were assessing zero-time lag interactions, as well as lagged interactions, between arrays. We did this because we felt that the interactions might take place on time-scales less than our 20 ms bin size. However, we have repeated this analysis by using only lagged values in one array to predict future values in the other array (i.e., in the new analysis l takes on values 1-10). The results were nearly identical (compare Fig. R1A1 and R1A2 and R1B1 and R1B2). We have included Fig. R1A2 and R1B2 in the supplementary material as Fig. 12.

$$p_i(\text{choice} | r_i(t)) = a_0 + \sum_{j=1:10} a_{(i,j)} p_i(\text{choice} | r_i(t-j)) + \sum_{k=1:8 \setminus i} \sum_{l=0:10} a_{(k,l)} p_k(\text{choice} | r_k(t-l))$$

Fig. R1. Comparison of information flow with and without zero-lag cross factor. A1. Original analysis, equivalent to Fig. 6D in the paper. B1 Same as A1 for prediction of object with direction. A2. Same as A1, except the analysis was carried out using $l = 1 : 10$ instead of $l = 0 : 10$. B2. Same as B1 with $l = 1 : 10$.

2. Related to the previous aspect, the directional information analysis includes time lag parameters, but the presentation of the results do not allow discerning a possible time lag of predictability of information in one array from the information that built up in the other arrays (partial model analysis and its difference to the full model). To reflect a directionality of prediction it is necessary to quantify and visualize how much information is there in the sender before it is available in the receiver.

Response: Thank you for this point. We do show one example set of time lagged parameter values in Fig. 4B. All of the coefficient values we assessed looked like this. Effectively, predictions were strongest at shorter lags, and then decayed with additional lags up to about 6 (i.e. 120 ms). We have now made this clearer in the results by adding: **“The model resulted in a set of kernel coefficients (Fig. 4B), which were convolved with the posteriors in the input arrays (Fig. 4C) to generate a prediction of the posterior on the output array (Fig. 4A). The kernel coefficients show the effect of lagged information in one area on future information in another area. This example shows that prediction tended to be strongest at short delays, and decay with time (Fig. 4B).”**

3. It is difficult to understand the effect size of the 'information' / posterior probability reflects. Given the decoding accuracy of 60-80% how much *_more_* information is conveyed by caudolateral choice related activity. Can the authors translate their measures into a more common measure (to learn how many choices are better predicted from caudolateral than from rostralateral arrays)?

Addressing this aspect might also help to understand why the author interpret the found gradients as "substantial" (line 521).

Response: The average of the posterior over trials will approximately equal the percent correct from decoding all trials. So the values in Fig. 3 are approximately the average posteriors. Fig. 5 shows the average posteriors (panels C-F), which increase from rostral to caudal, similar to the decoding accuracy shown in Fig. 3. The values in Panels G-J show the effect of dropping one array. The largest effects are around 1-2% in this example. Thus, the difference in the posterior (which is approximately the percent correct) is about 1-2% when we drop one array from the equation. Note that the effects of using the lagged values in one array to predict itself are always the largest (about 2%), but the effects of using another array can get to about ½ of that (i.e., 1%). Part of this has to do with the correlations among all arrays. Thus, these effects would be larger if we asked, how much better can we predict the posterior in array 4, when we use only array 3, not including other arrays. But we used the more statistically conservative approach of asking, if we include all arrays in the model, how much do we lose when we drop array 3. These are the values shown in Fig. 5 and Fig. S9, for the example of predicting actions in Where blocks.

We have removed reference to “substantial” when discussing information flow in the conclusion. We still include it when discussing the gradients, because this refers to the analyses in Fig. 2, which we feel do show a large difference in the fraction of neurons activated by the task factors, as well as the relative timing of that activation.

4. The study assumes and conveys explicitly that neurons in prefrontal cortex encode the identity of an object or a

spatial location, which is misleading. The analysis operationalizes the encoding of an object (or location) that is chosen (and fixated for 400ms). What is encoded is then the 'chosen object identity' and the 'chosen location'. This is different to conveying that an object identity is encoded. I am not sure whether the authors can analyze object identity as they always show the same two objects in a block and hence only the choice and the location of these objects vary. The task design does not allow distinguishing whether a neuron encodes within a block the rewarded object or the nonrewarded object (both are present on the screen), only whether the chosen object is differently encoded than the nonchosen object. This should be clarified to prevent readers to wrongly understand that objects or action are encoded independent of choices in these areas.

Response: Thank you for pointing this out. Our task does not allow us to investigate how the LPFC neurons encode object identity. It can only tell whether the neurons can discriminate the difference between chosen and non-chosen object identities (or locations). We clarified the expression to avoid misleading: “We began by characterizing single cell encoding of the chosen object and the chosen action. This analysis was split out by block type and anatomical location (Fig. 2; Supplementary Fig. 3). To be more specific, this analysis was performed to test whether and when single cells discriminated chosen and non-chosen options. This analysis does not specifically assess whether neurons encoded the object identity or location, only differences between chosen and unchosen actions or objects.”

5. The abstract states that the study "examined information timing and flow across". ... but the latencies of the information flow is not statically analyzed in depth.

Response: The comment about information timing in the abstract refers to the onset latencies shown in Figs. 2H and 2I and 3H and 3I, which we analyzed in detail. With respect to information flow, we hope we have clarified this point in response to comment 1.

6. The abstract lacks focus and is very general. Given the specific findings of a gradient can the abstract not introduce more explicitly the concepts of segregation (by domains) and integration (by functional requirement) and highlight the novel aspects of the results including also the fact that the information analysis allows to track learning the relevant objects ?

Response: We have modified the abstract by including the following, “The object to action effects were more pronounced in object blocks, and also reflected learning specifically in these blocks.”

7. It is not clear whether the main gradient results were evident similarly in each of the two monkeys, and in each hemisphere. How reproducible are these results?

Response: The main results were consistent among the two monkeys and hemispheres. We included comparisons between hemispheres in the supplemental materials. We plotted the raw population activity for each of the hemispheres in Fig. S2, the fraction of responsive neurons for each hemisphere in Fig. S3, decoding accuracy of chosen action/object and reward for each hemisphere in Fig. S6 and Fig. S8, and the information flow gradient for chosen action and object across domain type separately each monkey in Fig. S11. Although the effects were larger in one monkey for information flow, the trend was in the same direction.

8. Fig 6: the legend and axis labels are very difficult to understand. Please describe the axis meaning more explicitly in the legend.

Response: The legend has been updated as below:

Fig. 6. Prediction of chosen action and chosen object. (A-D) Contribution of arrays separated by different ordinal distances either rostral or caudal to each array to the prediction of posterior in each array. The x-axis indicates the ordinal distance (see Fig. 1D for array number) of the arrays which were dropped in the partial model. The y-axis indicates the difference of variance explained between the Partial models (i.e., when dropping one array, bilaterally) and the Full model (Δ Var), normalized by the variance explained in the Full model (fVar). Larger values indicate a stronger prediction of the posterior from the arrays separated by distances indicated on the x-axis. The separate panels show the prediction of decoding accuracy of action with action (A), predicting object with object (B), predicting action with object (C), and predicting object with action (D). "Rostral-Caudal" indicates information flow from rostral to caudal LPFC, labeled by negative ordinal distance; "Caudal-Rostral" indicates information flow from caudal to rostral LPFC, labeled by positive ordinal distance.

9. A key paper showing dorsal-ventral gradient of space/object coding in PFC that is not yet considered but of particular importance is:

Lebedev MA, Messinger A, Kralik JD, Wise SP (2004) Representation of attended versus remembered locations in prefrontal cortex. *PLoS Biol* 2: e365.

This reviewer has no ties to these authors.

Response: The suggested reference has been cited in the manuscript:

Neurons that code visual-spatial information are located in the cdIPFC, while those that code object identity information are located in the vlPFC^{7, 44, 45}.

10. The methods section about the decoding analysis does not describe what the input was for the decoder (single neuron data), how the neural data was normalized or not prior to decoding, what the decoding method actually was and how the data from the different monkeys were combined.

Response: In all cases we used the neurons collected in the same session to do the decoding analysis. So, the number of samples (N) equals 8 (2 monkeys \times 4 sessions) in Fig. 3. Also, the decoding method has been modified as:

We carried out the decoding analysis on chosen actions, chosen objects, and reward (i.e., reward or non-reward). Therefore, for these analyses, the chance performance was 50%. Analyses were carried out using leave one trial out cross-validation. The model was fit with the remaining trials and tested on the trial that was held out of the analysis. All simultaneously recorded neurons from each array were used to predict the indicated factor. Decoding was computed in 20 ms bins, advanced in 20 ms increments, time-locked to the chosen options on. Neural activity was not normalized or transformed. Raw spike counts in 20 ms bins for each neuron were used.

The posterior probability of choice, which is the probability of selecting the more rewarding action or object over trials, was calculated by:

$$p_i(t) = \frac{\exp(-(x_k(t) - \bar{X}_i(t))^2)}{\sum_{j=1:2} \exp(-(x_k(t) - \bar{X}_j(t))^2)}$$

Here $p_i(t)$ represents the choice probability for option i at time t , the vector $x_k(t)$ represents the neural population activity, with each element of the vector representing the spike count of a single neuron, in a single trial k , at time t . The vector $\bar{X}_i(t)$ represents the mean neural population activity across trials for a chosen object or action which was indicated by i or j . This is a linear decoder which, in a probabilistic sense would be a linear Gaussian decoder with a spherical covariance matrix.

11. Because neural response latencies are analyzed with differences in what and where blocks it seems important to know whether the initial behavioral response latencies in both blocks types were similar (were behavioral latencies also slower in the action blocks?).

Response: Those were not, as we mentioned in the main text, “We analyzed the reaction times (RTs) in both What and Where blocks. In What blocks, the average RT was 216.8 ms (SD = 12.3 ms), and in Where blocks, the average RT was 205.2 ms (SD = 19.6 ms). These RTs differed by block type (paired t-test, $t(7) = 4.08$, $p = 0.005$).”

The possible reason was also discussed, “In Where blocks, the animals did not have to use object information to select an action, they could simply pre-plan an action. The action was directed at an object. However, in What blocks, the animals had to use object information to find the object, and then direct a saccade towards it. Therefore, we expected information flow from object to action, but less flow from action to object (Supplementary Fig. 10).”

12. How did the authors use neurons recorded from the same array at successive days? Is it important to distinguish whether neurons on successive days are identical and thus should not be counted and analyzed twice?

Response: This is always an important question. We did not notice that neurons tended to be consistent across days on each electrode. But we did not explicitly assess this. To address this we used the data collected from one day (one session per day) for calculation of statistical significance ($n = 8$ sessions). Thus, we were examining consistency across sessions as opposed to using individual neurons as the statistical unit.

13. It would help to learn about the sorting criterion used to characterize action potentials to be from a single neuron. If the action potentials are from different neurons or are not separated from each other it is important to convey that the data are based on multiunit activities. Were effects dependent on the isolation quality of the neurons. This would simplify future studies aimed to reproduce the effects based on multiunit activity. In this context it is unclear what the average firing rate of the neurons were.

Response: The Method has been updated, “Single units were manually sorted offline. The threshold for spike acquisition was set at 4.5 x root mean square (RMS) of the baseline signal for each electrode.”

14. The methods section misses a description of the information flow analysis was statistically tested.

Response: Because we had a large amount of data, the information flow regression was always significant within a single session. However, in the results we focused on differences across conditions and caudal-rostral directions for the information flow analysis. This was tested using an unpaired t-test. As we mentioned in the main text, “We found that there was stronger flow in the caudo-rostral direction than in the rostral-caudal direction when predicting action with object (Fig. 6C; unpaired t-test, $t(766) = 7.39$, $p < 0.001$). We also found that there was stronger flow in the rostral-caudal direction than in the caudo-rostral direction when predicting object with action (Fig. 6D; unpaired t-test, $t(766) = 3.83$, $p < 0.001$).”

The idea was to compare the similarity of decoding among subregions in rostral-caudal and caudal-rostral directions.

15. The methods section has no information about the cross-spectral analysis. Which method, time window, statistics, etc. was used?

Response: Information about the cross-spectral analysis was included in the Supplementary Material. A section named *Local field potential (LFP) analysis* describes the methods, and a section named *Supplementary Results: LFP analysis* that describes the results in detail.

16. The study used arrays from both hemispheres but it is difficult to discern whether the main effects of the study were similarly evident in each hemisphere or whether the gradients and the encoding/decoding showed some hemisphere specificity.

Response: See reply to question 7 above.

17. In the results section it does not become clear why the decoding analysis was done after already quantifying encoding? An additional sentence motivating the analysis seems useful.

Response: We added a sentence to introduce the decoding analysis: “The encoding analysis addressed how individual neurons respond to chosen objects and actions. To further understand how the neural populations coded object and action information, we carried out a decoding analysis...”

Reviewer #3 (Remarks to the Author):

Review – Meso-scale functional organization of macaque lateral prefrontal cortex
Tang, Bartolo, and Averbeck

The manuscript by Tang and colleagues describes the findings of an heroic set of experiments in which the authors set out to address a question of long-standing interest with respect to the organization of the lateral prefrontal cortex – whether the representation of different types of task-relevant information are represented preferentially in specific LPFC subregions. To address this question, 4 Utah arrays were implanted bilaterally in two monkeys, each covering a specific LPFC subregion (rdLPFC, mdLPFC, cdLPFC, and vdLPFC). Monkeys performed a well-established task – a two-armed bandit task in which saccades were rewarded stochastically based on either the identity of a target object or the location to which the saccade was directed in a series of separate trial blocks. Population analyses of encoding, decoding, and information flow across arrays were conducted to determine whether representations of objects or actions were more prevalent in any PFC subregion, whether the latencies of these representations varied between regions, and how information regarding objects and actions flowed between different subregions. The authors conclude that:

- 1) more neurons are responsive during the task in caudal than rostral PFC
- 2) Both chosen objects and chosen actions are encoded in vdLPFC and cdLPFC, though a bias exists toward chosen objects in vdLPFC and chosen actions in cdLPFC
- 3) Response latencies of the population followed a caudal-rostral gradient, with latencies being shorter in caudal LPFC
- 4) Response latencies for reward information were similar across arrays.
- 5) Co-occurrence of action and object representations was stronger in caudal than rostral LPFC
- 6) Decoding performance for predicting chosen object or actions was greater for caudal than rostral LPFC. Similar caudal-rostral gradient of response latencies as encoding analysis.
- 7) Information flow was strongest between neighbouring arrays. Overall information flow was strongest in caudo-rostral direction when predicting actions.
- 8) Depending on task requirements, information regarding actions flows from dorsal to ventral LPFC, and information about objects flows from ventral to dorsal LPFC.

Overall, the authors conclude that several gradients exist in the caudo-rostral direction. Information carried by single neurons, as was as information flow is strongest in this direction. This is also the case for response latency, and information about both object identity and location are both present. A bias toward representing action information is present in cdLPFC, and a bias toward object information is present in vdLPFC. Information about action is transmitted from cdLPFC to vdLPFC when actions are relevant, and from vdLPFC to cdLPFC when object identity is relevant to the task at hand.

This is a thoughtfully designed and skillfully executed series of experiments. The experimental task is appropriate to address the questions at hand, the quality of the data is high, and all the analyses reported are sound. The conclusions flow correctly from the analyses as described. Although in one sense the reporting of mixed action and object selectivity across PFC subregions is expected from the large body of literature in this field, the authors have been able to go a step farther and directly address questions of information flow across PFC by recording from large samples of neurons in specific PFC subregions simultaneously. This represents a substantial advance and allows the novel conclusions generated by their analyses of information flow. I would recommend publication of this study. I have a few questions and comments, outlined below.

1) If I understand correctly, in their latency analyses, the authors report the latency at which, during the task, the percentage of task-related neurons exceeds that observed in the baseline period. Essentially my understanding here is that this represents the latency at which the population of neurons carrying task-relevant representations of object or action come "on-line" during the task. I wonder if a further analysis at the single-neuron level might shed more light on the differences between subregions here. Could the authors conduct an analysis to determine the latency at which single neurons discriminate the preferred from non-preferred object during the 'what' blocks, and the preferred from non-preferred action during the 'where' blocks? A sliding ROC or the sliding t-test type of analysis could be used to compare these two activity profiles and a discrimination time could be determined for each single unit and averaged across the different arrays to compare latency differences. Such an analysis might complement the existing population-level analysis done here.

Response: Apologies for not being clearer. If we understand the comment, we think we had done an analysis which addresses this. However, we may not have explained this clearly. The analysis is shown in Fig. 2. The latencies are shown in Fig. 2H for preferred actions and 2I for preferred objects. These are onset latencies computed across the population of single neurons, but with the analysis first done in each individual single neuron. For each single neuron, we assessed with an ANOVA, when it first discriminated chosen actions (i.e., when left vs. right was chosen) and chosen objects. In Fig. 2, we plot the fraction of single neurons that discriminated each factor as a function of time for each array. We then carried out a second analysis, as suggested, which asked when the population response became statistically significant. We used the population analysis to generate statistics for latency. But in fact, it was based on the analysis of when each single neuron became significant for each factor. Comparisons of the latencies for each array, for each condition, and for the choice of actions vs. objects are shown in Fig. 2H and 2I. We have clarified this in the results, which now states, "We began by characterizing single cell encoding of the chosen object and the chosen action. This analysis was split out by block type and anatomical location (Fig. 2; Supplementary Fig. 3). To be more specific, this analysis was performed to test whether and when single cells discriminated chosen and non-chosen options. This analysis does not specifically assess whether neurons encoded the object identity or location, only differences between chosen and unchosen actions or objects."

A 4-way ANOVA was then used to test the difference among regions. Details were described in Method: "To detect the response latencies, a paired t-test was performed between the average percentage of task-related neurons of the baseline period (from -1.5 to -0.5 seconds from the chosen options on) and each bin across the whole trial time course from all eight arrays. The first time point that showed a significant difference was defined as the response latency. A 4-way ANOVA was applied to examine the difference of response latencies from each region, which was carried out with leave one session out. The ANOVA was a standard, non-nested, linear model and with two levels of interactions. Factors included cerebral hemisphere, block type, domain type, and array locations."

The ANOVA and t-test results were indicated by * symbols in each panel in Fig. 2 and Fig. 3, as mentioned in the figure legends: "The black * symbols at the top of each panel indicate a significant difference among the four regions (1-way ANOVA, $p < 0.01$). The colored * symbols indicate a significant difference (paired t-test, $p < 0.01$) of task-related neuron percentage between the corresponding region and its baseline."

2) The authors report that the reaction times for the 'where' trials was significantly shorter than that for 'what' trials. However, in the latency analysis outlined, it is demonstrated that the latency for object representations is significantly shorter than that for actions. I wonder then at the behavioural relevance of the signals reported here. I wonder if a more direct analysis of the relationship between activity in the different PFC subregions and reaction times might add something here. For example, could the authors investigate the trial-by-trial correlation between activity at the end of the fixation period and the reaction times of the forthcoming saccades? Although, as the authors correctly point out, these signals are not explicitly "motor", we know that they are most likely being sent to areas that are in order to bias behavior appropriately. Also, given the seeming increasing abstraction of information represented along the caudo-rostral gradient, it might be expected that significant correlations would be obtained for the caudal but not rostral LPFC areas, thus showing further differences between regions.

Response: We carried out another analysis to address the correlation between single-unit activity and reaction times (see below). The neural activities were taken from the last 300 ms of the fixation period. We calculated the correlation in a trial-by-trial manner, and plotted the distribution of the r values of all single neurons recorded in each array, split by What and Where blocks. The distributions of What and Where blocks were similar to each other in all four regions (chi-square test, $P > 0.01$; in the rdIPFC, $M = 0.0009$, $SD = 0.0466$ for What blocks, $M = -0.0010$, $SD = 0.0430$ for Where blocks; in the mdIPFC, $M = 0.0026$, $SD = 0.0519$ for What blocks, $M = -0.0000$, $SD = 0.0440$ for Where blocks; in the cdIPFC, $M = -0.0021$, $SD = 0.0639$ for What blocks, $M = -0.0056$, $SD = 0.0517$ for Where blocks; in the vlIPFC, $M = -0.0125$, $SD = 0.0668$ for What blocks, $M = -0.0103$, $SD = 0.0568$ for Where blocks).

Some reasons may cause this phenomenon. First, although the reaction times in "where" blocks ($M = 205.2$ ms, $SD = 19.6$ ms) were significantly shorter than that in "what" blocks ($M = 216.8$ ms, $SD = 12.3$ ms), the absolute difference was small: about 11 ms. We may not be able to detect such a subtle difference in PFC neural activity. Second, some neurons may predict the

choices, but since the choices were close to the cue onsets, the neuronal activity responses to the choices mixed with the cue response causing it to be hard to detect.

3) Much, if not most of the electrophysiological evidence investigating domain-specificity in macaque PFC has been obtained by investigating the nature of persistent activity during delay periods of oculomotor delayed-response or delayed-match-to-sample tasks. Here the authors have used a different type of task and are investigating differences in activity during task epochs involving fixation before and after onset of choice stimuli in different conditions. I don't see this as any major issue, however, some acknowledgement of and comment on this would be appropriate in the discussion.

Response: We added sentences in the main text to address the difference between our task and the other delayed-response task: "Most previous electrophysiological data examining the role of LPFC in cognition was obtained using delayed-response tasks^{7, 11}. These tasks investigate the maintenance of "action" or "object" information over time. Our task was designed to address how the LPFC neurons dissociate the "action" and "object" information during rapid learning from reinforcement. We focused our analyses on the anatomical information flow between domains."

REVIEWERS' COMMENTS

Reviewer #1 (Remarks to the Author):

The revised submission has improved greatly in clarity.

I think the addition of one sentence and new reference may help strengthen the overall point regarding gradients in cortical function.

In the following paragraph:

"Furthermore, when we examined the caudal-rostral representation of block type (i.e., What or Where), which is an abstract rule that defines the relevant learning dimension, we did not find an enriched representation more rostrally. We did not find any factors that dominated in the rdIPFC, although reward showed less of a gradient along this axis, with no difference in response latency. It is possible that if we had used a task with a different form of abstraction, we would have engaged the rdIPFC more strongly."

Add this example to emphasize the point: ^[1]_[5P]

"For example, neurons that evaluate self-performance have been found in the rhesus monkey frontal pole, consistent with higher order, metacognitive abilities residing in more rostral locations (Tsujimoto et al., Trends Cogn Sci. 15: 169-76, 2011)."

Reviewer #2 (Remarks to the Author):

The authors constructively and convincingly addressed the main concerns. This is an impressive, sophisticated paper that promises to have significant impact on the field.

Reviewer #3 (Remarks to the Author):

The authors have more than adequately addressed my comments and I am happy to recommend this manuscript for publication.

We are grateful to the reviewers for their careful reading of our manuscript, their insightful comments, and their overall positive evaluation. We have made extensive changes to address all issues raised by the referees (*in italics below*).

Reviewer #1 (Remarks to the Author):

The revised submission has improved greatly in clarity.

I think the addition of one sentence and new reference may help strengthen the overall point regarding gradients in cortical function.

In the following paragraph:

"Furthermore, when we examined the caudal-rostral representation of block type (i.e., What or Where), which is an abstract rule that defines the relevant learning dimension, we did not find an enriched representation more rostrally. We did not find any factors that dominated in the rdIPFC, although reward showed less of a gradient along this axis, with no difference in response latency. It is possible that if we had used a task with a different form of abstraction, we would have engaged the rdIPFC more strongly."

Add this example to emphasize the point: □

"For example, neurons that evaluate self-performance have been found in the rhesus monkey frontal pole, consistent with higher order, metacognitive abilities residing in more rostral locations (Tsujimoto et al., Trends Cogn Sci. 15: 169-76, 2011)."

Response: The suggested reference has been cited in the manuscript:

It is possible that if we had used a task with a different form of abstraction, we would have engaged the rdIPFC more strongly. For example, neurons that evaluate self-performance have been found in the rhesus monkey frontal pole, consistent with higher order, metacognitive abilities residing in more rostral locations ⁶⁶.